# Deciphering key processes controlling rainfall isotopic variability during extreme tropical cyclones

Ricardo Sánchez-Murillo [1]*, Ana M. Durán-Quesada[2], Germain Esquivel-Hernández[1], Daniela Rojas-Cantillano[3], Christian Birkel[4,5], Kristen Welsh[6], Minerva Sánchez-Llull[7], Carlos M. Alonso-Hernández[7], Doerthe Tetzlaff [8,9,5], Chris Soulsby[5], Jan Boll[10], Naoyuki Kurita[11] & Kim M. Cobb[12]

The Mesoamerican and Caribbean (MAC) region is characterized by tropical cyclones (TCs), strong El Niño-Southern Oscillation events, and climate variability that bring unique hazards to socio-ecological systems. Here we report the first characterization of the isotopic evolution of a TC (Hurricane Otto, 2016) in the MAC region. We use long-term daily rainfall isotopes from Costa Rica and event-based sampling of Hurricanes Irma and Maria (2017), to underpin the dynamical drivers of TC isotope ratios. During Hurricane Otto, rainfall exhibited a large isotopic range, comparable to the annual isotopic cycle. As Hurricane Otto organized into a Category 3, rapid isotopic depletion coupled with a decrease in $d$-excess indicates efficient isotopic fractionation within ~200 km SW of the warm core. Our results shed light on key processes governing rainfall isotope ratios in the MAC region during continental and maritime TC tracks, with applications to the interpretation of paleo-hydroclimate across the tropics.

[1] Stable Isotope Research Group, Universidad Nacional, Heredia 86-3000, Costa Rica. [2] Atmospheric, Oceanic and Planetary Physics and Climate System Observation Laboratory, School of Physics and Center of Geophysical Research, University of Costa Rica, San José 11501-2060, Costa Rica. [3] HIDROCEC, Universidad Nacional, Liberia 50101, Costa Rica. [4] Department of Geography, University of Costa Rica, San José 2060, Costa Rica. [5] Northern Rivers Institute, University of Aberdeen, Aberdeen AB24 3UE, UK. [6] School of Chemistry, Environmental and Life Sciences, University of The Bahamas, Nassau N-4912, Bahamas. [7] Centro de Estudios Ambientales de Cienfuegos, Cienfuegos AP 5, Cuba. [8] Department of Geography, Humboldt University, Berlin, Berlin 12489, Germany. [9] IGB Leibniz Institute of Freshwater Ecology and Inland Fisheries, Berlin 12587, Germany. [10] Civil and Environmental Engineering, Washington State University, Pullman, WA 99164-2910, USA. [11] Graduate School of Environmental Studies, Nagoya University, Nagoya 464-8601, Japan. [12] School of Earth and Atmospheric Sciences, Georgia Institute of Technology, Atlanta, GA 30332-0340, USA. *email: ricardo.sanchez.murillo@una.cr

   

In recent years, the Mesoamerican and Caribbean (MAC) region has experienced an increase in the frequency and intensity of extreme weather events. More intense and frequent tropical cyclones (TCs) as well as large interannual rainfall variability result in extensive human, ecological, and economic damage[1–7], highlighting the need for an improved understanding of past and present-day TC dynamics, and how they might evolve under continued greenhouse forcing. Projections suggest TCs will likely strengthen under continued warming, with a two- to threefold increase in frequency of Category 4 (H4) and 5 (H5) hurricanes in the Atlantic basin between 20° and 40°N by 2050 (ref. [8]). It has proven difficult to detect significant trends in recent TC statistics owing to the relatively short instrumental record of TC activity[9–17]. Paleoclimate reconstructions of TC activity are crucial to quantifying the full range of TC variability—both forced and unforced—that are key to the assessments of natural hazard risk management in the tropics[18–24]. Such reconstructions often rely on oxygen isotope variations in geological archives[9], often in combination with isotope-enable models[25], all of which rely on relatively scarce modern-day observations of rainfall isotopes.

Modern TC rainfall sampling is essential to interpret the intensity and frequency of extreme periods recorded in terrestrial and maritime paleo-archives (e.g., caves, corals, lake sediments, and potentially tree-rings)[9]. Depleted calcite-$\delta^{18}O$ records within the MAC region have commonly been hypothesized to be the net result of concerted forcing by the North Atlantic Oscillation (NAO), the Atlantic Multidecadal Oscillation (AMO), and anomalous shifting of the Intertropical Convergence Zone (ITCZ)[9–17]. In conjugation, these large-scale processes are theorized to result in a strong isotopic amount effect[26] (a low latitude inverse correlation between the $\delta^{18}O$ and the amount of rainfall). Enriched calcite-$\delta^{18}O$ excursions are often considered to be a direct effect of long-term droughts, which facilitate the occurrence of below cloud-base kinetic fractionation during short lived and less intense rainfall events and subsequent enrichment near the surface and in the vadose zone[9–16] during infiltration. Furthermore, qualitatively past rainfall interpretations largely rely on the empirical slope of a modern amount effect as a transfer function[27] to infer the occurrence of wet and dry periods in the tropics. Typically, this function is estimated by linear regression analysis between monthly or more long-term mean rainfall and isotopic values available from historical monitoring stations (i.e., Global Network of Isotopes in Precipitation, GNIP hereafter). As GNIP sites are often several hundred kilometers away from the study site and represent discontinuous time series, relying on the assumption that local rainfall producing conditions and orographic effects (if present) are to some degree similar to the area of interest, it may fail to capture processes affecting regions featured by large spatial heterogeneity. In the absence of recent or historical isotopic records, regionalized amount effect relationships are assumed valid in order to classify wet and dry episodes, which provide an incomplete perspective of tropical rainfall processes.

Water isotopologues provide key insights into the complex processes related to the development and dissipation of TCs and mid-latitude cyclones[28–32]. A recent study[33] demonstrated that bulk rainfall microphysics and cloud type exert comparable pan-tropical influence in the observed isotope variability of rainfall, with moisture transport playing a secondary role in regions characterized by deep convective development (e.g., MAC region). While correlations between monthly scale rainfall $\delta^{18}O$ and rainfall amount are fairly high across the tropics, such analyses mask fundamental processes that occur on shorter timescales (e.g., moisture convergence and entrainment). Indeed, weak correlations between daily rainfall $\delta^{18}O$ and rainfall amount across the tropics underscore that >80% of the daily rainfall $\delta^{18}O$ variance is unexplained by the amount effect[33]. In TC-active regions, a complete understanding of the dominant hydrological processes that contribute to past, present, and future hydroclimate variability must include $\delta^{18}O$ observations at synoptic spatial and temporal scales.

In this study, we examine the anatomy of Hurricane Otto—a large TC event in the MAC region—with respect to water isotopologues ($\delta^{18}O$, $\delta^{2}H$, and $d$-excess) and associated hydrometeorological data. We compare our results to those from two large Atlantic TCs (Hurricanes Irma and Maria) in September 2017, as well as to other tropical and mid-latitude cyclones (both continental and maritime landfalls). Daily resolved and TC-related isotopic observations demonstrate that enriched rainfall isotopic ratios occur during both large and small rainfall events within the central MAC region. Across this region, off-shore convection can result in large and enriched rainfall amounts during TC passages and may potentially bias such reconstructions in favor of heavy water isotopes, typically linked to drier conditions. We also demonstrate that the relationship between precipitation amount (type) and isotopic composition varies spatially and temporally, echoing the importance of process-based studies of modern-day rainfall isotope distributions on daily to interannual timescales at paleoclimate reconstruction sites.

## Results

**Hurricane Otto synoptic characteristics.** Aside from its unique trajectory (Fig. 1a), Hurricane Otto broke several records for TCs in the MAC region. When Otto reached hurricane status at 1800 UTC (Universal Time Coordinated) November 23, it exceeded Hurricane Martha (1969) as the latest and strongest TC of the year, becoming the latest hurricane on record in the Caribbean basin. Hurricane Otto's landfall was also the southernmost recorded in Central America, surpassing Hurricane Irene (1971)[2]. Unlike most North Atlantic TCs, Hurricane Otto featured a particular southeasterly origin and trajectory (compared to over 100 years of major hurricane tracks across the eastern Pacific Ocean, Caribbean Sea, and North Atlantic Ocean basins) followed by landfall about 18.5 km north of the Nicaragua-Costa Rica border as a Category 3 hurricane (H3)[34] (Fig. 1b). According to modern records, Hurricane Otto was the only known TC to move over Costa Rica and was the first Atlantic TC to cross into the eastern Pacific Ocean as a TC since Hurricane Cesar in 1996. Hurricane Otto was responsible for 18 casualties within Costa Rica and Panama[2].

Historically, TC impacts on Costa Rica were restricted to indirect effects on precipitation causing moderate to large rainfall accumulation[35] (Fig. 1a). A well-dated stalagmite from Belize (i.e., 450-year reconstruction with annual resolution) indicates that TC activity within the western Caribbean Sea basin peaked around 1650 A.D. and decreased gradually until 1983 (ref. [17]). Thus, recent TC activity and the projected changes may present a major shift in hurricane patterns within the MAC region from less to more active seasons.

The convective disturbances that gave rise to Hurricane Otto are linked to interactions between convectively coupled Kelvin wave and tropical waves passing through the southwestern Caribbean Sea in November 2016. After disorganized cloudiness resulted from a low pressure system and the passage of tropical waves, the low-level vorticity was fed by increasing northerly low-level winds and high sea surface temperatures (SST > 29 °C). A Tropical Depression (TD)[34] developed by November 20 near 1800 UTC with an eastward drifting path. The cyclone strengthened to a Tropical Storm (TS)[34] by November 21 0600 UTC (Fig. 2a). At this stage, organized

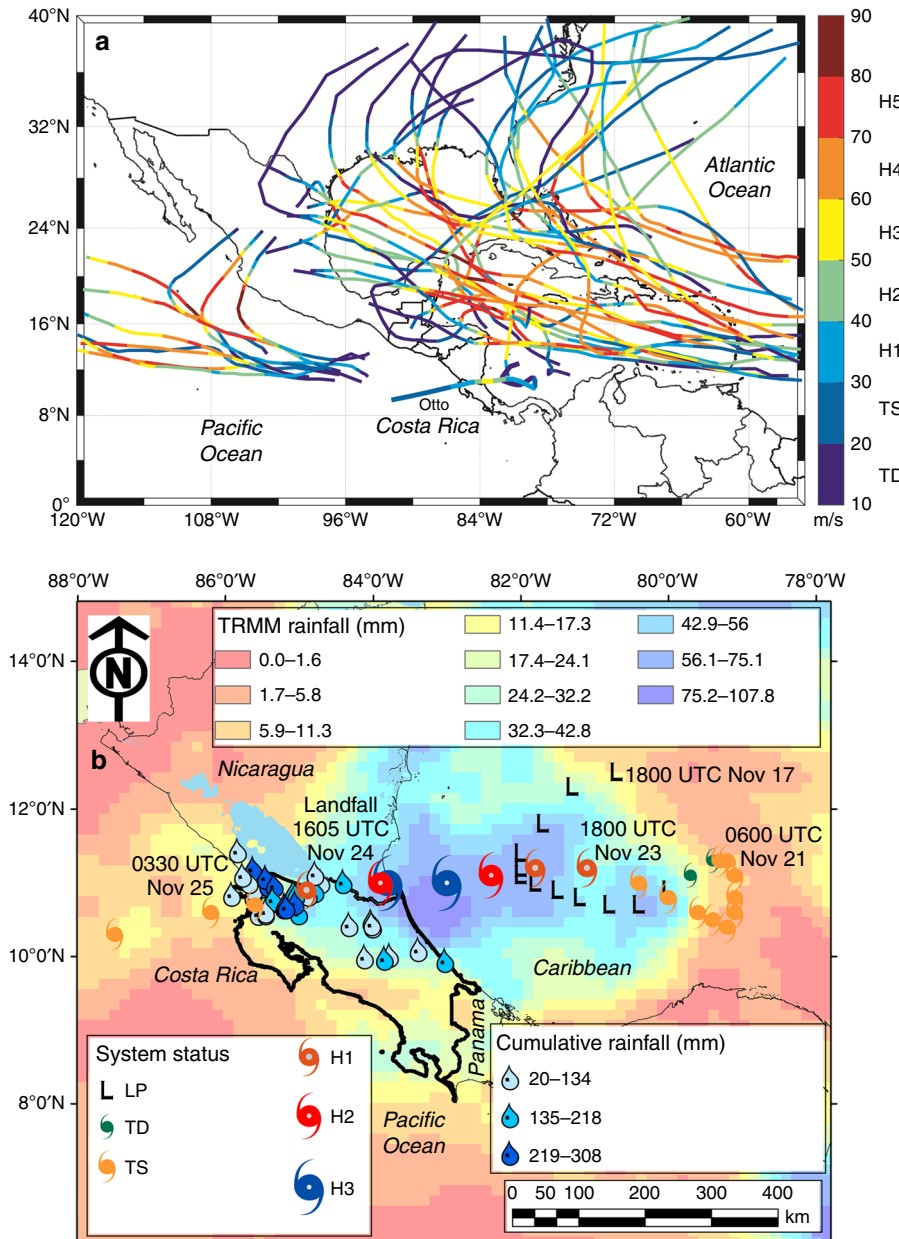

**Fig. 1** Historical, major hurricane tracks comparison and Hurricane Otto main characteristics. **a** Historical, category 5 hurricane tracks with at least one segment along trajectory as category 3 for the Atlantic and eastern Tropical Pacific basins (1900–2017), bolder trajectory line denotes the best track of Hurricane Otto (November 20–26, 2016). Historical trajectories were constructed based on a combined dataset of NHC (National Hurricane Center) archive and HURDAT (Hurricane Databases) (https://www.aoml.noaa.gov/hrd/hurdat/Data_Storm.html). Wind speed (m/s) segments are color-coded with their respective storm category. **b** System status from 1800 UTC (Universal Time Coordinated) November 18 to 1200 UTC November 25, 2016. Raindrops represent ground-based cumulative rainfall during November 24–25, 2016. Raster color-coded grid shows cumulative rainfall estimated for November 23–25, 2016 from TRMM (Tropical Rainfall Measuring Mission) (https://trmm.gsfc.nasa.gov/)

convection featured two convective nuclei (shown as 1 and 2 in Fig. 2a) over the inner Caribbean Sea and southernmost Costa Rican Pacific coast, respectively. The wave disturbance supported the increase in rainfall over the Costa Rican southern region, preceding the intensification of the slow moving cyclone. After November 23, 1815 UTC, deep convection organized around the tropical cyclone center (black open circle in Fig. 2b), with the moist air being pulled in, forming a well-defined cyclonic band. Rainfall associated with this moving band had a significant effect on the accumulated precipitation over Costa Rica and southern Nicaragua prior to the hurricane intensification (Fig. 1b).

As it moved westwards across the Caribbean, Otto developed into a Category 1 (H1)[34] storm with a convective band over the southernmost Central American Caribbean coast. By 1215 UTC November 24, Otto exhibited a well-defined eye-wall structure with the low pressure center approaching the Costa Rica-Nicaragua border (Fig. 2c). Intense low-level winds enhanced the interaction with the topography and added a local enhancement to the increasing energy availability for convective activation. Otto made landfall at approximately 1605 UTC November 24, as H3, followed by a rapid weakening to a Category 2 (H2)[34] (Fig. 2d). The hurricane continued its trajectory to the west over the northern Costa Rican lowlands

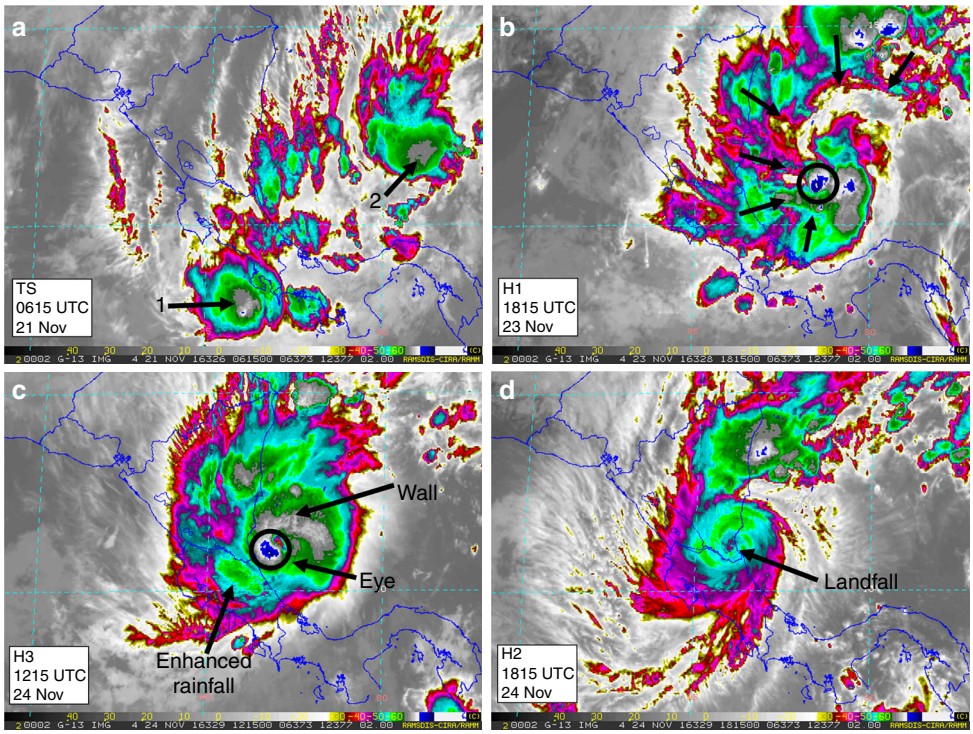

**Fig. 2** Satellite evolution of Hurricane Otto. GOES-8 (Geostationary Operational Environmental Satellite) East Thermal IR brightness temperature (IRTB) imagery retrieved from the RAMSDIS (Regional Advanced Meteorological Satellite Demonstration and Interpretation System) website during the event (http://rammb.cira.colostate.edu/ramsdis/online/). The evolution of Hurricane Otto is shown as the organization of convection going from a TD (Tropical Depression) to TS (Tropical Storm) (**a**) (two convective nuclei are noted by numbers 1 and 2). Well-defined convective clouds (shielded by the green IRTB contours) organized around the low pressure minimum (**b**) displaying a cyclonic motion as the system intensified to a H1 (black arrows denote dragged moisture transport). Favorable conditions, including warm water (SST > 29 °C) allowed the increase to H3 with a well-defined and nearly steady eye and wall (**c**). Green contours over northeastern Costa Rica show evidence of convective rainfall as Otto approached land (1605 UTC November 24, 2016) (**d**). Despite the rapid decline of the hurricane, its pass across the Costa Rica-Nicaragua border was followed by high convective rainfall rates

with a nearly steady eye. Heavy rainfall along its northern passage, most-likely enhanced by moisture supply from Lake Nicaragua (Fig. 1b; lake area = 8264 km²), its growing proximity to the Pacific Ocean and the passage of the hurricane over a highly deforested area, was associated with severe flooding and damage along the northern lowlands and inter-mountainous border areas (Fig. 1b), with 6-h cumulative rainfall amounts of up to 308 mm. The intensity of the cyclone rapidly diminished, and Otto reached the eastern Pacific Ocean as TS at around 0330 UTC November 25 (Fig. 1b).

**Rainfall isotope variability within the MAC region.** In the central MAC region, current seasonality from dry (Dec–Apr) to wet season (May–Nov) is characterized by two notable V- or W-shaped isotopic patterns[36–38] in rainfall (Fig. 3a). Spectral analysis of the $\delta^{18}O$ time series exhibits a dominant rainfall cycle ranging from 60 to 15 days, with maximum peak activities between 41 and 22 days (Fig. 3b; Supplementary Data 3–4). These patterns are consistent with the intra-seasonal variability of Central America rainfall that typically result in two or three depleted excursions during the wet season and two enriched pulses during the mid-summer drought (MSD; ITCZ northward migration ~8°N and development of deep convection over the Caribbean domain between July and August)[39] and the months of the strongest trade winds (Jan–Feb). For instance, by mid-May when the ITCZ passes over the MAC region, a first sharp depletion in the isotope composition is commonly observed (Fig. 3a). Slight enrichment occurs during the transition to the MSD. Later in September and October, the isotopic composition shows a second strong depletion, often accompanied by the indirect influence of TCs. As the

ITCZ migrates southward by mid-November, a second enrichment is observed towards mid-December (Fig. 3a). These patterns are amplified within the Pacific slope of the MAC region, whereas in the Caribbean domain (i.e., juxtaposed coasts and inner Caribbean Sea islands) the isotopic composition is less variable throughout the year[38].

Positive (enriched) and negative (depleted) excursions in the isotopic composition of tropical rainfall have been linked to the occurrence of dry and wet conditions, respectively[10–15]. These types of statistical inferences are often not physically based, and interpretations are even more problematic as the sampling frequency increases (i.e., weekly to sub-daily)[33]. Figure 4a shows the regression between the stratiform rainfall area fraction ($F_{st}$) and the area-averaged rainfall amount near the Caribbean coast of Costa Rica (Adj. $R^2 = 0.30$, $P < 0.001$) (Supplementary Data 5). In the tropics, rainfall represents the combination of convective (deep and shallow) and stratiform rainfall fractions. Strong convective cells (i.e., strong updrafts > 10 km) with potentially shorter moisture recycling times result in more enriched isotope compositions, whereas larger stratiform area fractions are represented by more depleted values (Fig. 4b). Recent findings across the globe provide additional evidence of such a mechanism[27,33,40–42]. The large degree of scatter as the stratiform area fraction increases may be attributed to the combination of stratiform rainfall and shallow convection during rainfall events.

**Isotope anatomy of a major hurricane in the MAC region.** The influence of Hurricane Otto lasted seven days over Costa Rica, Nicaragua, and Panama, resulting in a large spectrum of isotopic compositions from its genesis and development in the southern

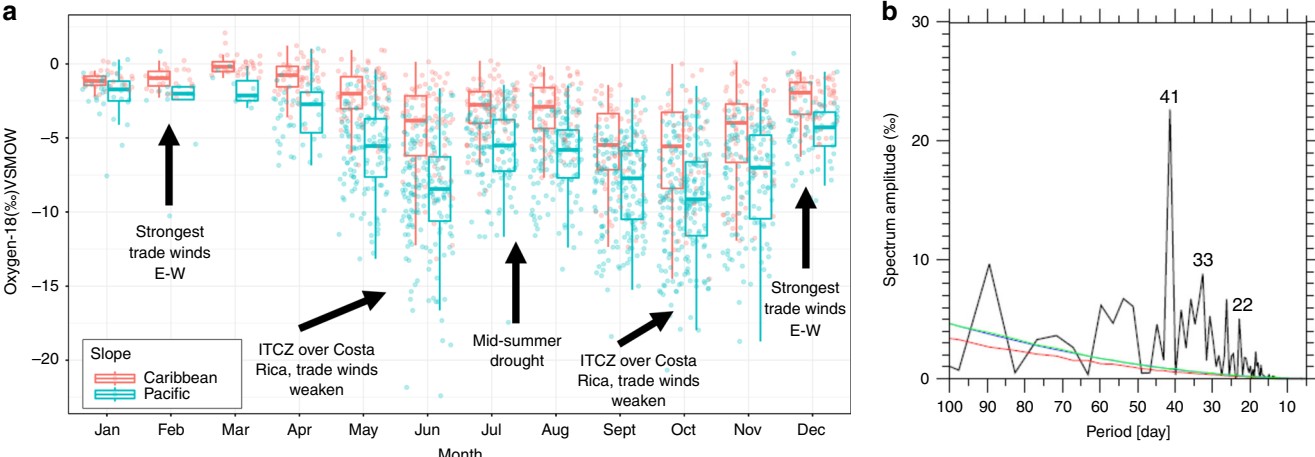

**Fig. 3** Stable isotope variability ($\delta^{18}O$, ‰) between 2013 and 2017 from the stable isotope monitoring network in Costa Rica. **a** Caribbean and Pacific slopes are denoted by red and cyan colors, respectively. Black arrows represent three main climatic features: east-west trade winds, displacement of the ITCZ (Intertropical Convergence Zone), and the MSD (Mid-summer drought). **b** Power spectrum of $\delta^{18}O$ (spectrum amplitude, ‰) with Red Noise spectrum and 90% confidence interval (red and green lines, respectively) generated using REDFIT. The numbers denote the most significant frequency patterns

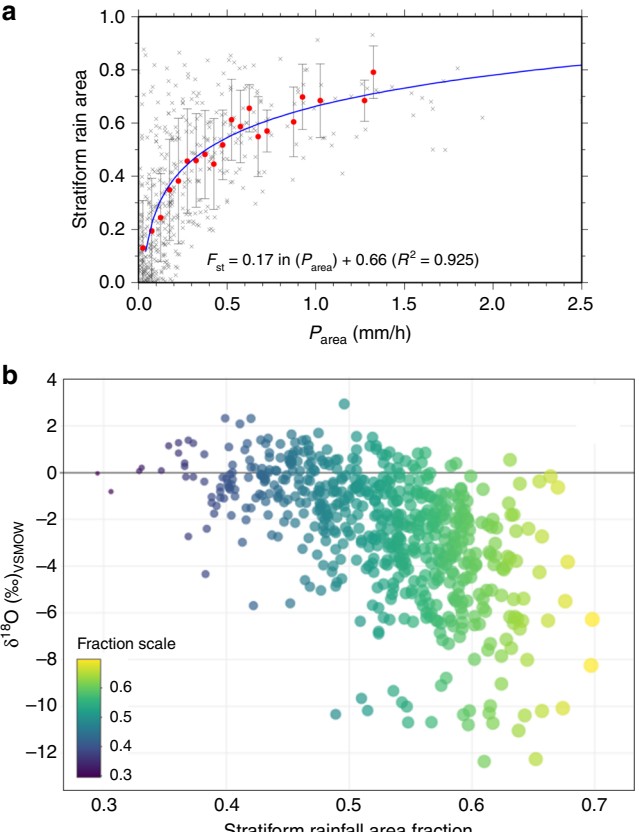

**Fig. 4** Stratiform rainfall area fraction and $\delta^{18}O$ (‰) relationship for a long-term monitoring station in the Caribbean coast of Costa Rica during 2013–2017. **a** Relationship between stratiform rainfall area fraction ($F_{st}$) and the area-averaged rainfall amount over the 5° × 5° longitude/latitude box centered at the 28 Millas station during the observation period (March 2014–September 2017). Red circles with error bars represent the average and standard deviation in precipitation intensity bins for each 0.05 mm/h interval up to 1.5 mm/h. Blue curved line shows a logarithmic regression of averaged values. **b** Stratiform rainfall area fraction and stable isotope composition $\delta^{18}O$ relationship (Adj. $R^2 = 0.30$; $P < 0.001$). Stratiform rainfall area fraction is color and size coded. The horizontal black line represents 0.0 (‰) in $\delta^{18}O$ as a reference

Caribbean Sea to its dissipation over the eastern Pacific Ocean[2] (Fig. 5a–e; Supplementary Data 1). Less organized stages such as TD and TS produced large isotopic variability ranging from +0.10‰ up to −12.39‰ in $\delta^{18}O$ (Figs. 5f and 6a). The system presented two stages as category H1 (Fig. 1b). The first stage was associated with convective activity primarily concentrated ~210 km from the Caribbean coast of Costa Rica and Nicaragua, which produced enriched values up to +1.89‰ in $\delta^{18}O$ (Fig. 5f; maritime-influenced rainfall). The second H1 stage developed when Hurricane Otto moved across the northern mountainous range of Costa Rica, resulting in more depleted isotope ratios (up to −6.22‰ in $\delta^{18}O$; more continental interaction) (Fig. 5f). Our monitoring captured the expected enrichment linked with off-shore convection (maritime TC signal), confirming the typical indirect effect TCs have on the coastal region and the contrasting large depletion observed over continental land rarely measured during TCs within the MAC region (Fig. 6a).

As a category H2, Otto's isotopic footprint evolved through two different stages. When the system reached H2 in front of the Costa Rica and Nicaragua Caribbean coast (~142 km from the coast), isotope composition decreased to −4.91‰ in $\delta^{18}O$. After landfall as H3, and the subsequent decrease to H2 within the landmass area, isotope composition showed a depletion to −10.24‰ in $\delta^{18}O$ (Fig. 5f). Landfall as category H3 resulted in the most depleted rainfall with values ranging from −9.36 to −21.06‰ in $\delta^{18}O$ (Fig. 5f). The isotopic composition of Otto as a H3 agrees with previously reported tropical (eastern and western Pacific Ocean) and sub-tropical (north Atlantic Ocean) cyclone compositions (Fig. 6a)[28–32]. Recent isotopic monitoring during Hurricanes Irma and Maria (both category H5 within the MAC region in 2017) revealed maritime-type enriched compositions (from ~0‰ up to −5‰ in $\delta^{18}O$). The genesis, development, and dissipation of these two large TC systems were concentrated within the Caribbean-Atlantic basin, maintaining strong convective activity along the track. Interestingly, these category H5 events did not result in depleted compositions at Cuba or The Bahamas (Supplementary Data 2). Given that most of the significant paleo-archives (e.g., Yucatán Peninsula[43]) of the MAC region are located along the Caribbean coast, the observed isotopic differences between the indirect versus direct effect of TCs is key to accurate interpretations of such records. Our results confirm that in this region, off-shore convection can result in large and enriched rainfall amounts during TC passages and may

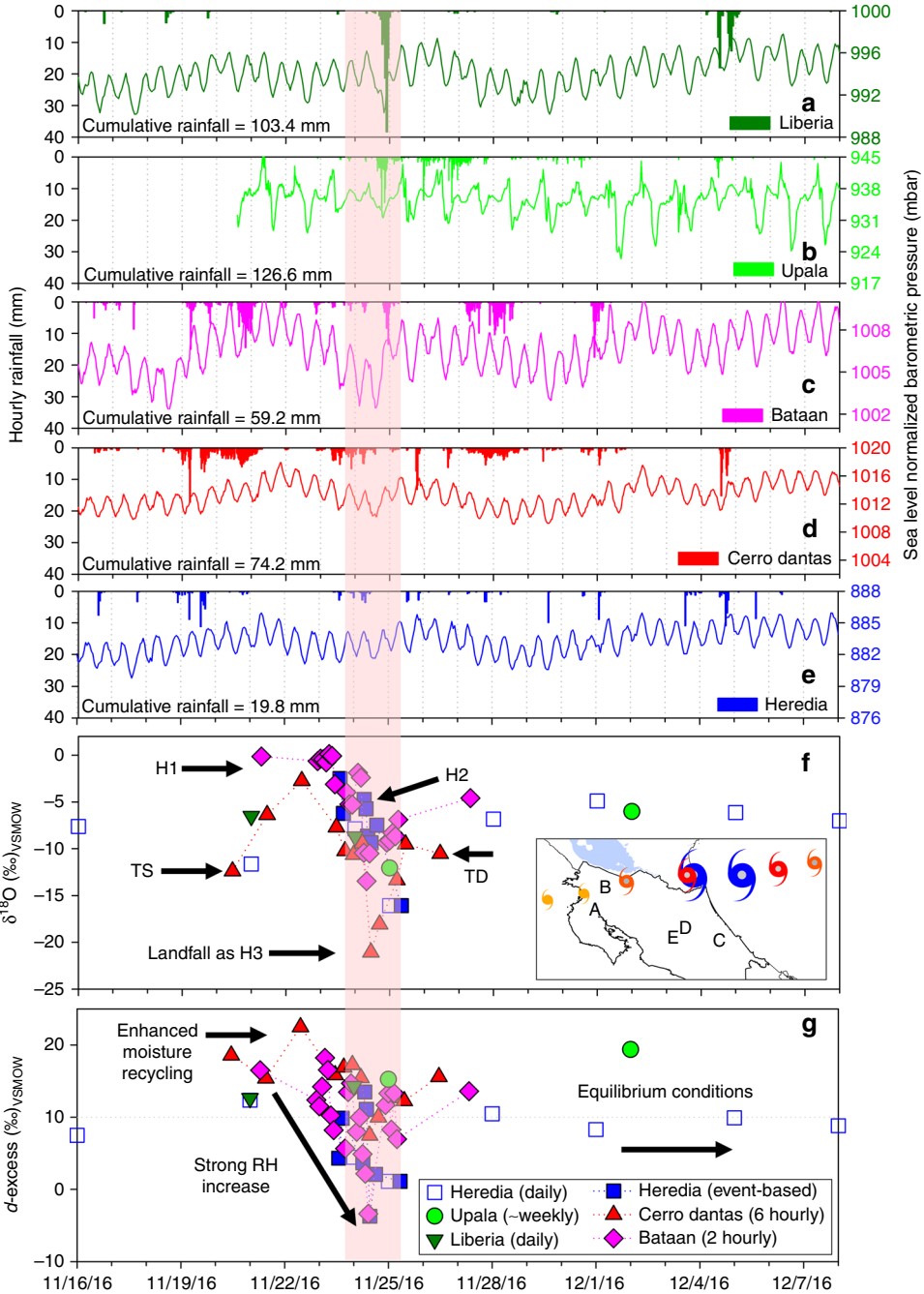

**Fig. 5** Isotopic and meteorological co-evolution of Hurricane Otto at five selected locations (color-coded) across the Pacific and Caribbean slopes of Costa Rica. **a–e** Hourly rainfall (mm) and sea level normalized barometric pressure (mbar). Cumulative rainfall values correspond to November 24–25, 2016. **f, g** Present the $\delta^{18}O$ (‰) and d-excess (‰) variability prior to, during, and after Hurricane Otto passage across the Costa Rica-Nicaragua border at event-based, 2-hourly, 6-hourly, daily, and weekly sampling frequencies, respectively. Map inset in **f** shows the location of the selected sites relative to the hurricane trajectory

potentially bias such reconstructions in favor of heavy water isotopes, which researchers may erroneously link to drier conditions via the amount effect.

Rainfall d-excess values during the Hurricane Otto displayed appreciable variability that largely covaried with rainfall $\delta^{18}O$, ranging from −9.4 to +22.5‰ (Fig. 5g). Increases in relative humidity (near saturation) and wind speed (~185 km h⁻¹) as the system gained organization towards category H3 (~975 mb) in the inner Caribbean Sea were reflected in low surface d-excess values (−9.4‰)[31,44], whereas potentially continental moisture

recycling resulted in large d-excess values up to +22.5‰[36–38]. The isotopic east-west spatiotemporal structure of Otto are related to near-equilibrium maritime d-excess values during genesis and early development, low values during the maximum strength of the hurricane, and high d-excess values towards the dissipation of the cyclone (Fig. 6b). The isotopic anatomy of Otto also provides detail of the convective to stratiform ratio that featured the passage of the hurricane across the Costa Rica-Nicaragua border. Enriched samples were retrieved from the periods in which the TC exhibited its maximum intensity along

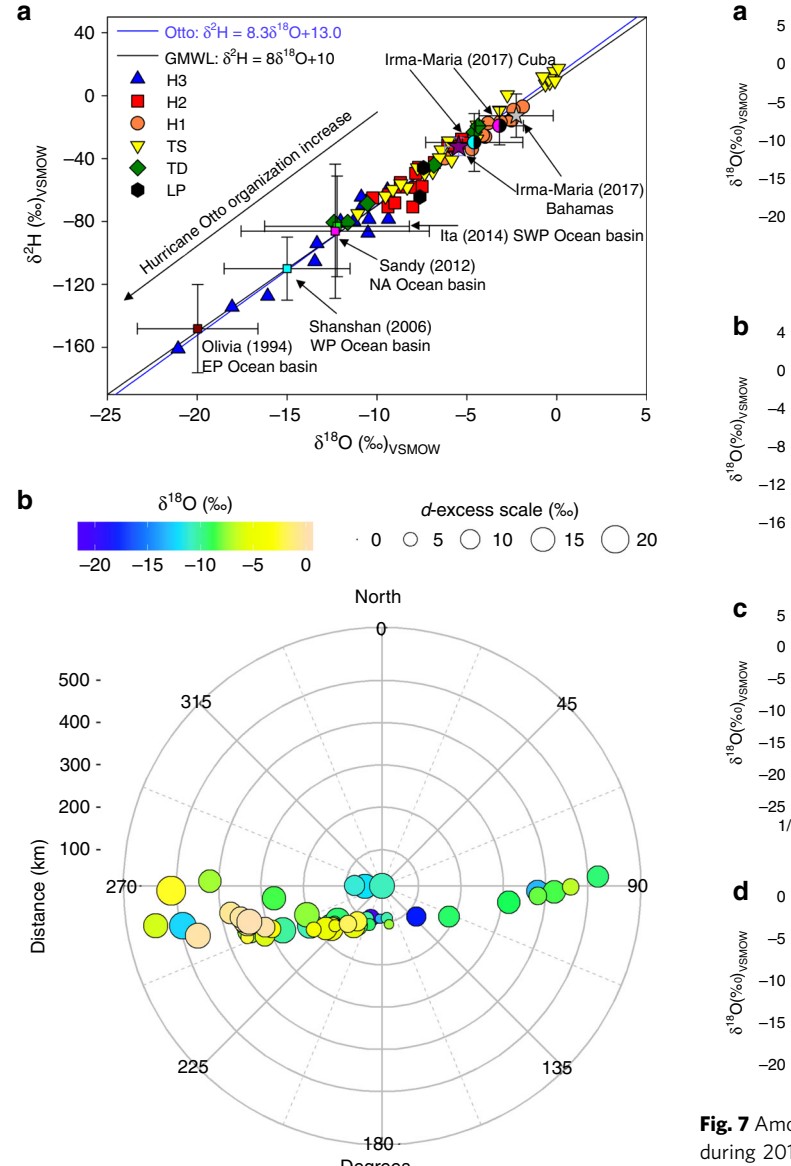

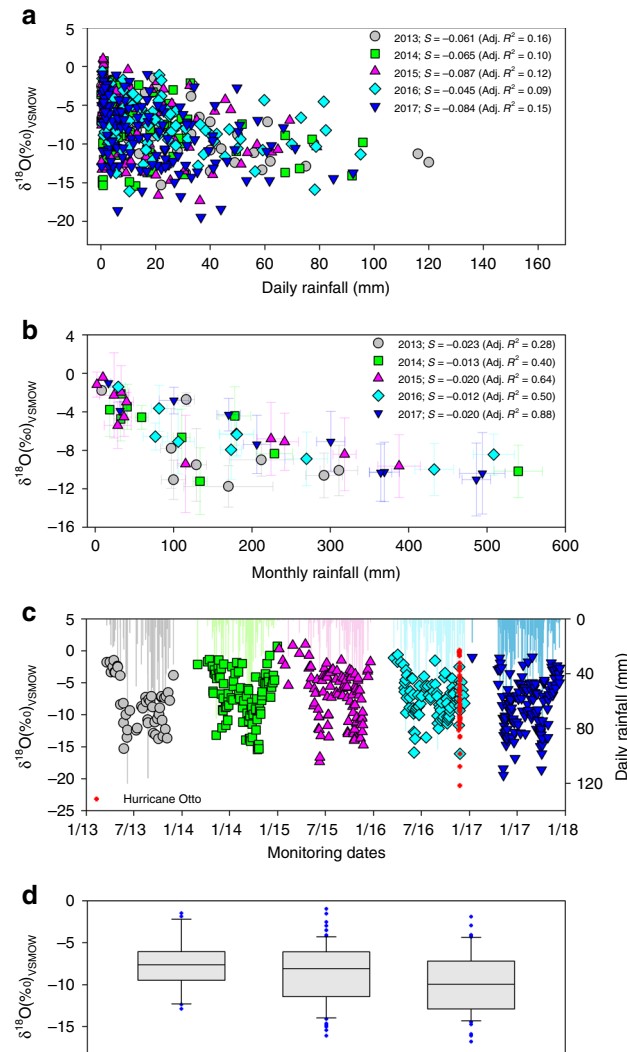

**Fig. 6** Major tropical cyclones (TCs) isotopic comparison and evolution. **a** Dual isotope diagram showing the isotopic evolution of Hurricane Otto compared to six large cyclones: Olivia (1994, Eastern Pacific Ocean, never touched land), Shanshan (2006, Western Pacific Ocean), Sandy (2012, North Atlantic Ocean), Ita (2014, Southwestern Pacific Ocean), Irma (2017, Atlantic Ocean), and Maria (2017, Atlantic Ocean). Hurricane Otto's movement from the semi-closed Caribbean Sea basin to the eastern Pacific Ocean revealed a large isotopic spectrum in the cyclone anatomy. The gray and purple stars represent the average isotope composition during the indirect influence of TCs Irma and Maria near Nassau, The Bahamas, respectively. The light blue and pink semi-filled circles denote the average isotope composition during the passage of TCs Irma and Maria near Cienfuegos, Cuba, respectively. **b** Isotopic evolution in space relative to the cyclone center. $\delta^{18}O$ (‰) variability is color-coded and d-excess (‰) changes are denoted by the circle size scale. Concentric gray circles represent the estimated distance (km) to the storm center

the maritime trajectory and the coldest cloud top shields were observed (convective activity in the cumulonimbus tower). In contrast, depleted samples coincide with decaying mature clouds cells featured by stratiform less intense rainfall as observed in the band structure of the TC across the continental land.

**Fig. 7** Amount effect and rainfall $\delta^{18}O$ (‰) time series in central Costa Rica during 2013–2017. **a** Daily mean $\delta^{18}O$ (‰) versus rainfall amount (mm). **b** Monthly mean $\delta^{18}O$ (‰) versus rainfall amount (mm). Amount effect slopes (S) and adjusted $R^2$ (Adj. $R^2$) are included for each year. **c** Daily rainfall $\delta^{18}O$ (‰) variability during 2013–2017 and Hurricane Otto's isotopic values (red crosses). Color-coded bars denote daily rainfall in mm. **d** Box-plots of isotope composition in Costa Rica under the indirect influence of hurricanes (H1, H2, H3), major hurricanes (H4, H5), and tropical storms (TS)

## Discussion

High-frequency water sampling across a large spatial scale during Hurricane Otto's passage over Costa Rica provided novel evidence for both depleted and enriched rainfall isotopic variability throughout the storm's evolution, respectively. Such data represent a window into mesoscale convective processes during TC events, which can aid the development of model simulations of shallow to deep convection in such weather extremes.

Analyses of daily rainfall $\delta^{18}O$ observations (2013–2017) suggest that the amount effect can only explain 9–16% of the total variance (Fig. 7a). When aggregated to monthly means, the amount effect explains 28–88% of the variance observed in monthly isotopic composition[33] (Fig. 7b). While the amount effect slope (S) of 2016 exhibited a significant reduction in comparison with the longer monitoring period,

there is no such difference when slopes are computed using monthly means.

Our daily resolved and TC-related isotopic observations demonstrate that enriched isotopic ratios occur during both large and small rainfall events within the central MAC region (Fig. 7c). Similarly, depleted isotopic ratios can also occur during both large and small rainfall events. However, during the study interval, tropical storms were the most common origin of significantly depleted isotopic ratios, whereas the farther-field indirect effect of hurricanes was represented by more enriched compositions (Fig. 7d).

Event-based observations from tropical stations over land[45] and ocean[46] identified organized convective systems as the dominant control on rainfall isotope variability in the MAC region. One study attributed the amount effect to the post-condensational processes in the unsaturated downdrafts[47] through the environmental subsidence and mixing during the organized convective events. Recent observational studies[27,40,42] found that the isotope ratios in tropical and mid-latitude precipitation reflect the proportions of convective/stratiform rains. These findings lead to the fundamental question of whether the amount effect across the tropics is more related to changes in the total amount of precipitation and the net changes in convective activity and the types of precipitation at a region, both of which may evolve under continued anthropogenic climate change.

Our isotopic evidence contributes to the assessment of the relative importance of the amount effect and other atmospheric processes (e.g., moisture origin, maritime versus terrestrial influences, and rainfall type) on isotope distributions on daily to interannual timescales. Apart from their paleoclimate applications, such detailed datasets are increasingly the subject of data-model inter-comparisons that serve as key diagnostic windows on model parameterizations of mesoscale convective development[27,33]. Similarly, our study has several key implications for the interpretation of paleoclimate reconstructions of tropical hydrology based on water stable isotopes. First, we demonstrate that the relationship between precipitation amount (type) and isotopic composition varies spatially and temporally, echoing the importance of process-based studies of modern-day rainfall isotope distributions on daily to interannual timescales at paleoclimate reconstruction sites. Second, based on our observations during Hurricane Otto, we hypothesize that in the MAC region, proxy records may be sensitive to the degree of convective versus stratiform activity in the past, in addition to the total amount of precipitation. Lastly, the observation of both highly depleted and enriched rainfall during Hurricane Otto may complicate the detection of TCs in even the most high-resolution archives of past rainfall isotope variations, supporting the development of a network of high-resolution proxy records for paleo-tempestology applications.

In the MAC region, our rainfall isotope observations provide a much-needed benchmark for the interpretation of paleoclimate reconstructions of rainfall isotopes. Most of the significant paleo-archives of the MAC region are located along the Caribbean coast. Across this region, off-shore convection can result in large and enriched rainfall amounts during TC passages and may potentially bias such reconstructions in favor of heavy water isotopes, typically linked to drier conditions. Therefore, we urge the paleoclimate community to consider the full range of climatic phenomenon that contribute to rainfall $\delta^{18}O$ variability at their site, and in this sense our manuscript provides quantitative constraints on a weather extreme that plays a key role in climate across the MAC region. As a critical next step, simultaneous sampling of surface and subsurface water during TC passages would lead to better understanding of how TCs impact both paleoclimatic archives as well as regional hydrology, both of which have important implications for water resource management under continued climate change[48–55].

## Methods

**TC satellite imagery and cumulative rainfall calculations.** To provide context for Hurricane Otto's trajectory within past TCs, historical H5 tracks with at least one segment along the trajectory as H3 for the Atlantic Ocean and eastern Tropical Pacific basins (1900–2017), were reconstructed based on a combined dataset of NOAA's National Hurricane Center archive and Hurricane Research Division data[56]. Total rainfall amounts for November 23–25, 2016 were calculated from the Experimental Real-Time TRMM Multi-Satellite Precipitation Analysis data (http://disc.gsfc.nasa.gov/datacollection/3B4XRT_V7.shtml) with a 0.25° latitude and longitude and 3-hourly spatiotemporal resolution. Gridded data resulted from the combination of merging microwave, microwave-calibrated infrared, and combined microwave-infrared estimates of precipitation.

From the 3-hourly gridded data, daily rainfall totals were estimated[57] and were used to generate a geostatistical interpolation applying an Empirical Bayesian Kriging (EBK) method in ArcGIS 10.4 (ESRI, USA). Unlike other kriging methods, which use weighted least squares, the semivariogram parameters in EBK are estimated using restricted maximum likelihood. The Thin Plate Spline semivariogram for a given distance $h$ is reproduced by $\gamma(h) = \text{Nugget} + b|h^2| \cdot ln(|h|)$, where the Nugget and slope $b$ must be positive[58]. Satellite imagery from GOES-8 East 2 km IR4 was used to analyze the evolution of the system based on cloud top temperature, air mass boundary movements, convergence margins, and presence of thunderstorm activity (http://rammb.cira.colostate.edu/ramsdis/online/goes-west_goes-east.asp).

Complementary information on the moisture content of the water column was evaluated from selected GPS meteorology sites provided by CocoNet during Otto's evolution[59]. Ground-based cumulative rainfall (mm) and sea level normalized barometric pressure (mbar) in selected locations from November 24–25, 2016 were obtained from the Drought Observatory Monitor of the National University of Costa Rica and from the NOAA National Hurricane Center.

**TCs rainfall sampling and stable isotope analysis.** Rainfall samples from Hurricane Otto ($N = 74$, Supplementary Data 1) were collected using passive devices[60] across the Pacific and Caribbean domains of Costa Rica at different locations (as part of the Stable Isotope Monitoring Network of Costa Rica) covering an area of 51,100 km$^2$ and discrete time intervals (i.e., event-based; bi-hourly, 6-hourly, daily, and weekly) prior to, during, and after landfall (1605 UTC November 24, 2016). Samples were immediately transferred and stored in airtight 30 mL borosilicate containers at 5°C until analysis. Hurricane samples were analyzed at the Northern Rivers Institute at the University of Aberdeen using a Los Gatos DLT-100 laser analyzer with a precision of ±0.6‰ for $\delta^2H$ and ±0.1‰ for $\delta^{18}O$ (1 σ). Stable isotope abundances are expressed as $\delta^{18}O$ or $\delta^2H = (R_s/R_{std} - 1) \cdot 1000$, where $R$ is the $^{18}O/^{16}O$ or $^2H/^1H$ ratio in a sample (s) or standard (std) and reported in the delta-notation (‰) relative to Vienna Standard Mean Ocean Water (V-SMOW) reference standards. Instrument accuracy was assessed with a combination of in-house and external water standards (SMOW and SLAP). Deuterium excess was calculated as $d\text{-excess} = \delta^2H - 8 \cdot \delta^{18}O$[26].

Lately, more attention has been paid to $d$-excess change in a wide variety of paleoclimatic archives, such as ice cores, speleothems, and lake sediments in order to re-construct source precipitation conditions, coupled with subsequent analysis of potential air mass back trajectories[61–66]. Deuterium excess values may deviate from +10 (‰) due to the combination of three factors: a) relative humidity (RH) increase in the precipitation source, b) a decrease in SST, and c) greater wind speeds (>7 ms$^{-1}$) affecting evaporation and subsequent kinetic fractionation[44]. Thus, Hurricane Otto's $d$-excess evolution may provide new evidence to characterize tropical extreme events.

To complement the interpretation of Hurricane Otto's isotope composition, additional rainfall samples from the MAC region (Nassau, The Bahamas and Cienfuegos, Cuba) were collected using passive collectors during the passage of Hurricanes Irma and Maria in September 2017 ($N = 46$, Supplementary Data 2). After each rainfall event, samples were immediately transferred and stored in airtight 30 mL borosilicate containers at 5 °C until their shipping and analysis at the Stable Isotope Research Group facilities of the National University of Costa Rica. TCs isotope ratios within the MAC region were compared to previously published isotopic ratios from large tropical and mid-latitude cyclones, including both continental and maritime landfalls[28–32].

To our knowledge, this dataset is the first ground-based daily isotope record of rainfall within the MAC region covering the above average TC seasons (2016–2017), while also capturing all phases (warm, cold, and neutral) of the El Niño-Southern Oscillation (ENSO) in the last decade.

**Long-term daily rainfall and stable isotope analysis.** Hurricane Otto's isotopic response was analyzed in the context of a daily rainfall monitoring program ($N = 459$, 2013–2017, Supplementary Data 3) in central Costa Rica[36–38]. Samples were collected using a passive collector[60] and analyzed for isotopes at the Stable Isotope Research Group facilities of the National University of Costa Rica using a L2120-$i$ (Picarro, California, USA) and Los Gatos Research IWA-45EP (Los Gatos

Research, California, USA) water isotope laser analyzers. The analytical long-term uncertainty was: ±0.5 (‰) (1σ) for δ2H, ±0.1 (‰) (1σ) for δ18O. Linear regressions of rainfall amount versus δ18O (‰) (daily and monthly basis) were computed to investigate the effect of a TC isotope signal in the annual amount effect slope (S). The indirect effect of hurricanes (H1, H2, and H3), major hurricanes (H4 and H5), and tropical storms developed in the Caribbean basin was also analyzed in the framework of the isotopic variability of the MAC region (2013–2017).

**Spectral and stratiform rainfall analysis**. To understand the stable isotope interannual variability, a spectral analysis was carried out for daily isotope samples collected near the Caribbean coast of Costa Rica (28 Millas station, $N = 654$, 2014–2017, Supplementary Data 4). The REDFIT program was used to compute the Fourier analysis and confidence level of the dataset[67]. The significance of the REDFIT power spectrum was assessed by the red-noise spectrum using Monte Carlo simulation analysis. To examine the influence of the stratiform rainfall on daily isotopic variability, we computed the stratiform rainfall area fraction ($F_{st}$) near the Caribbean coast of Costa Rica. This study used the Ku-band Precipitation Radar (KuPR) convective/stratiform classification data from version 5 level 2 product of GPM (Global Precipitation Measurement) Core Observatory (https://pmm.nasa.gov/GPM), which is a successor of TRMM Precipitation Radar. $F_{st}$ is defined as the percent of total rain area covered by stratiform rainfall. Thus, we calculated stratiform rainfall area averaged over the 5° by 5° longitude/latitude box centered at the 28 Millas station, and then was divided by total rain area.

## Data availability

The authors declare that all data supporting the findings of this study are available in the article and in the Supplementary Data 1–5. Additional information is available from the corresponding author upon request.

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

## Acknowledgements

This study was supported by International Atomic Energy Agency grant CRP-19747 to R. S.M. under the pan-tropical initiative "Stable isotopes in precipitation and paleoclimatic archives in tropical areas to improve regional hydrological and climatic impact models". Analytical instrumental support from the IAEA Technical Cooperation Project (COS7005: Ensuring water security and sustainability of Costa Rica) is also acknowledged. Support from the Research Office of the Universidad Nacional of Costa Rica through grants SIA-0482–13, SIA-0378–14, and SIA-0101–14 was also fundamental. We thank various helping hands that contributed to rainfall sampling during Hurricanes Otto, Irma, and Maria, particularly to the personnel of the Estación Biológica 28 Millas (Bataan, Costa Rica), Cerro Dantas Refuge (Heredia, Costa Rica), Centro de Estudios Ambientales de Cienfuegos (Cienfuegos, Cuba), and the School of Chemistry at the University of The Bahamas (Nassau, The Bahamas). Support from the Isotope Network for Tropical Ecosystem Studies (ISONet) funded by the University of Costa Rica Research Council is also acknowledged.

## Author contributions

R.S.-M. conceived and designed the study in discussion with G.E.-H and D.R.-C. R.S.-M. performed the isotopic analysis. A.M.D.-Q. collaborated with satellite imagery analysis, meteorological, and dynamics interpretations. K.W., M.S.-L., and C.M.A.-H. collaborated with the isotope sampling of Hurricanes Irma and Maria in The Bahamas and Cuba, respectively. Stratiform fractions were computed by N.K. Tropical paleoclimatic framework was provided by K.M.C. R.S.-M., A.M.D.-Q., D.R.-C., C.B., K.W., C.M.A.-H., D.T., C.S., J.B., N.K., and K.M.C contributed to figure preparation, results interpretation, discussion of the associated hurricane isotopic dynamics, and paleoclimate interpretations and final preparation of the manuscript.

## Competing interests

The authors declare no competing interests.
