## [Peer Review File · Nature Communications]

Reviewers' comments:

Reviewer #1 (Remarks to the Author):

This paper by Sanchez-Murillo and colleagues presents a really valuable dataset on precipitation, surface water, and groundwater isotope ratios from Costa Rica. In addition to a long-term record, the authors conducted intensive sampling to characterize precipitation isotopes during the passage of a hurricane. This is of potential importance given the interest in using isotopes to trace the impact of severe storms on water resources and in reconstructing past storm activity using isotopic proxy data. The records are unique, but I find the analysis and interpretation in the manuscript, as submitted, lacking. On the one hand, the authors present a number of interpretations, many emphasized as major conclusions (e.g., convective vs stratiform precipitation as a control on storm isotope ratios, estimates of uncertainty in paleo-proxy interpretations using the amount effect, and dismissal of moisture source as an isotopic control, all of which appear in the abstract), that are not at all supported by the analyses presented here. Conversely, I feel that they miss several important opportunities to analyze their data in ways that could really speak to the issue of paleoclimate reconstruction signals, emphasized in the title but given second-shrift in the paper. These issues are described in more detail below.

I 335: I find most of the discussion in this section to be poorly supported. My impression is that the authors are overlooking the basic first-order patterns in their data and instead rushing to an overly complex and inadequately justified interpretation framed in terms of a current 'hot' idea (stratiform/convective precipitation). To first order, the main pattern I see in the hurricane timeseries is one that has been shown in datasets from other such storms (e.g., Olivia, Sandy), which is high values corresponding to the outer rain bands and dropping values as 1) the storm's eye approaches and more 'processed' moisture from inner rain bands is delivered, and 2) the storm's circulation begins to interact more strongly with land and less with the ocean. This can largely be explained in terms of upwind rainout and Rayleigh theory (Good et al., 2014). The roles of different moisture sources and stratiform/convective fractions are also worth considering, certainly, but the manuscript does not present any analyses that directly address these issues. I discuss concerns with the dismissal of vapor source controls below, and here just would point out that no data quantifying convective vs. stratiform precipitation are presented. This discussion section makes it sound as if the paper establishes these relations as truth, but no analyses are provided to back them up!

I 360-363: Not clear how increased humidity and wind speed produce negative d-excess values. I think you can get to zero this way but not below. What is the proposed mechanism here?

I 414: What is the basis for the dismissal of moisture source as a control on isotope ratios here? The referenced figure (6) provides no such analysis, and the text provides no justification. The subsequent reference to a comparison between Costa Rica and Vietnam seasonal patterns is trivial – although clearly these sites in different parts of the world receive moisture from different parts of the ocean there could be a common seasonal pattern in moisture source conditions (and isotope ratios). I'm not saying that I think this is true, just pointing out that the manuscript provides no concrete evidence against it. You would need to conduct moisture source attribution analysis to actually test the idea.

Figure 6: The comparison shown in panel B is very interesting...are the statistical distributions of Otto-associated d18O and the rest of the record at all different? It sure doesn't look like it. Nonetheless the surface and groundwater data show a spike, so I'm guessing that the amount-weighted rainfall for the storm looks very different than the unweighted mean of these fixed interval samples (why not present this comparison?). One could ask the question 'what would it take to actually detect this event (Otto) in a paleo-record', which is a very direct question that is at the heart of this manuscript but never actually addressed. The authors have an outstanding dataset to work with (both the precip and the sw and gw) and I would love to see them design some numerical experiments to really test this idea (e.g., simulate attenuation in different proxies types and ask whether an Otto signal stands out above the noise). IMHO this would be a more significant contribution than the tact the manuscript currently takes.

I 486-488: Is this true given the high degree of time-averaging (attenuation) of the paleo archive signals? I don't think so.

I488-492: As mentioned above, this has not been supported

Good, S. P., Mallia, D. V., Denis, E. H., Freeman, K. H., Feng, X., Li, S., . . . Bowen, G. J. (2014). High frequency trends in the isotopic composition of Superstorm Sandy. In J. B. Bennington & E. C. Farmer (Eds.), *Learning from the Impacts of Superstorm Sandy* (pp. 41-55): Academic Press.

Reviewer #2 (Remarks to the Author):

"Hurricane isotope anatomy in a climate change sensitive region: implications for tropical paleoclimatic re-constructions" presents a comprehensive new water isotope dataset from Costa Rica of daily precipitation sampled at multiple sites across the country as well as spring and surface waters. The dataset has captured a rare, southern hurricane path, and the authors combine precipitation and isotope data to describe the expression of the storm in isotope-space, and to examine the assumption that the tropical 'amount effect' holds on daily/event timescales, and what this all means in terms of interpreting isotope-based proxy data, commonly used to reconstruct past tropical cyclones and hurricanes.

The dataset is really fantastic and figures are also quite attractive and high quality. The authors have put a lot of work into this ms.

The main take home message I am getting is, as in many sites and recent papers, the classical tropical 'amount effect' does not hold on the event timescale in Costa Rica, generally, and when considering this hurricane. This is fine with me, but I think the authors could go further with their analysis of this dataset.

For example, In the second to last paragraph, the authors conclude that what is driving isotopic variability is "convective activity and ... fractions of shallow convective and stratiform rainfall generated, suggesting a major role of the in-cloud microphysics and thermodynamics (i.e., latent heat profile) within the precipitable water column, particularly between the convective and stratiform region in Mesoscale systems. Yet, there are no figures or analysis of % stratiform vs convective precip vs isotope values, OLR, or other meteorologic variables vs precip isotope values, so I don't see how this conclusion is drawn.

My biggest issue with the manuscript as it is written is the framework in which the data are presented and analyzed. The science problem the authors pose in the introduction is testing the 'amount effect' (and the moisture source effect, although they don't expand upon this as much) in order to properly interpret isotope-based paleoclimate proxies. However, the article fails to come 'full circle' and bring the reader back to any specific, or radical re-interpretation of proxy records. It's also unclear how many proxy records there are, and how many need to be reinterpreted etc., and what the implications of those re-interpretations may be. I suggest either 1) digging deeply into the paleodata or proxy system modeling to show how your results have clear impacts on interpretations, or 2) reframing the paper to focus more on understanding the dynamics and processes that lead to the observed isotope patterns. In that light, when introducing the amount effect, it is important to acknowledge the multiple hypotheses (many supported by observations) proposed in papers over the last decade regarding what produces the amount effect. The authors have a great vehicle in this dataset in which to systematically test some of these hypotheses.

Some other comments:

How many isotope based paleoclimate records from your region actually convert proxy d18O in mm/day space? The authors discuss this transfer function, but I wonder how often d18O is really converted.

I don't see the value in comparing this record from Costa Rica to a record from Vietnam, it does

not add much to the manuscript. The conclusion you are trying to make (about moisture source impact on precip $d_{18}O$?) is not coming across.

In terms of the authors concluding that moisture source is not important, I am struck by panel A in figure 2, which shows 2 disturbances, one in the Pacific, and one in the Caribbean. What about teasing out Pacific vs Caribbean moisture contributions to this event with some mixing models? And there is brief mention of a Nicaraguan lake moisture source contribution, but this is not analyzed further?

A major conclusion is the imprint of a single extreme event on surface and subsurface water (SF2). That is impressive and somewhat glossed over. These data are only talked about in terms of damping, but what about considering lag times, and the magnitude of isotope depletion required to be imprinted on surface and subsurface water? Building a pseudo-speleothem with a proxy system model (these are out there and not too complicated)? This is the information that directly translates to speleothem records.

The discussion of the range of isotope values in this storm suggests that a pdf type analysis, looking at tails (skewness) could also be really useful.

Author response for manuscript NCOMMS-18-01461A

General comments by the authors in response to criticisms raised by the Reviewers:

1. It is important to point out that our analysis characterizes the isotope composition of rainfall and surface/groundwater from a rare landfalling hurricane in Costa Rica. We produce evidence of the structure and behavior of this extraordinary event.
2. Previously reported evidence on the rain – isotope ratio relationship is mostly based on monthly samples, which do not necessarily capture details of relatively short lived and intense events unless during the sample collection period the event stands out compared to the rest of the rainy days. The heavy rainfall events accompanying a hurricane, such as in our case of hurricane Otto, can only be captured by high-resolution monitoring and sampling (at least daily). Moreover, there is a mixture of rain producing systems other than tropical cyclones, so that the signal is even more complex to define. In this study, the full signal of a hurricane event was captured in isolation since no other type of system was active at the same time. Hence, the record reported in this study allowed for a fully detailed analysis of the structure of the hurricane life cycle from genesis to dissipation in the context of long-term rainfall isotope monitoring.
3. Despite relatively similar results compared to previous analysis of hurricanes, the frequency provided in the present record guarantees that the life cycle changes of the system were properly tracked. Moreover, the variations in the isotope composition could be associated with a) changes in the latent heat release that feature the different stages of the hurricane, b) wind speed (as an inverse relationship of the minimum pressure) and rainfall intensity, c) location of the rain band and relative distance between the sampling site and eyewall and d) convective stability in order to improve the understanding of the hurricane evolution and rainfall distribution.
4. The abrupt decrease in the ^{18}O values of rainfall in fact respond to the processing of moisture in terms of forced changes due to the effect of the minimum pressure point. The pressure is reduced in the center of the system airflow and undergoes an adiabatic expansion and cooling, favoring the cloud formation process which organizes as part of the rain bands. As the clouds develop with distance from the low pressure center, established cloud droplets lead to raindrop formation initiating the rainfall process.
5. Upwind rainout explanation following Rayleigh theory may be a good approximation for larger areas. However, in the case of hurricane Otto, the shift between ocean and land interaction differs significantly from typical Atlantic hurricanes making landfall across the Gulf of Mexico, since the land portion is rather small. Interactions with land were fast so that the upwind rainout should be similar to that observed for orographic rainfall.
6. Moisture origin is not a relevant issue in this work as the Caribbean isotope signature source is well known. Even when the air masses undergo changes, the moisture origin for the systems remains the same. Changes in the isotope composition reveal variations in the elevation, saturation and movement of the air masses.
7. Changes due to the variations in specific humidity and wind speed that result from the mixing during the evolution of a TC, cause major variations of d -excess, and negative values of the d -excess are supported by the principles outlined and going back to

Daansgard (1964). The proposed mechanism is based on a fast rate of evaporation that occurs at non-equilibrium conditions as the moist airflow moves following the convergence to the low pressure center. As air moves to the low pressure center and expands, evaporation occurs, cloud droplets form supporting the organization of cloud formation which by continuity leads to the development of the rain bands. Deep clouds formed from these droplets at a high altitude and already represented by a low d -excess. The d -excess becomes lower as condensation occurs at a higher altitude, so that heavy rainfall from this type of system shows negative or low d -excess values. Hence, two conditions are associated with negative d -excess for this case: a) high evaporation rate as the air converges to the low-pressure system and b) high altitude condensation related to heavy rainfall from the rain bands.

8. Costa Rica is mainly receiving moisture from the Caribbean during the activation of the TC season. Therefore, the attribution of moisture sources is not necessary and has been well documented previously with different types of analysis.
9. The detection of an event like Otto from a paleo record-perspective would be a valuable contribution, however, there are some aspects that need to be considered:
 - Speleothems in the region are limited to the northern Caribbean, Belize, and Yucatán.
 - Note that rainfall in Yucatán is mostly driven by TC activity, so anomalously low ^{18}O values can be clearly identified as major TC events since other processes hardly occur. For Belize, the effect of heavy rainfall due to the ITCZ is masked and strong deep convection is not as active as it is for southern Central America. Northern records are not representative of the region and the analysis of TC for regions such as Costa Rica hold a greater complexity that requires a way to distinguish e.g. the isotopic spike for a heavy TC and the low isotopic value associated with deep convective systems due to other processes such as ITCZ-transient interaction.
 - Previous analysis with speleothems have provided limited interpretation of TC activity mostly directed to instrumental period variability, partly due to the masking issue, which makes it difficult to separate the signal to create an event based record. In order to design an experiment to evaluate whether Otto's signal or similar events can stand out, a signal value range attribution following a previous evaluation of other rain producing systems is required in order to make sure that the signals can be separated (e.g., ITCZ, MCS, easterly wave activity). This implies that a sort of catalog linking the present day known dynamic properties of the systems (spatial and temporal scales, rain intensity, cloud liquid/ice content, propagation speed) with the isotopic fingerprint must be created. From this perspective, this study contributes to such an effort by developing the database for further re-interpretation of paleo-records based on quantitative properties of the rain producing systems.
10. We present a hurricane isotope anatomy to shed light on the processes controlling the life cycle of a TC. Moist airflow that converges into the low pressure system that initiated Otto can be considered as featured by an enriched ^{18}O composition that results from the system being fed by moisture of Caribbean origin. Moreover, the origin of the moisture associated with rainfall over Costa Rica during hurricane Otto is not considered as a key concern in the manuscript, since the origin of moisture for the region has been well documented. In addition, we aimed to propose a more comprehensive explanation of how the variations in the isotope composition of rainfall samples along the hurricane life cycle are controlled by relying on an alternative to the traditional amount effect.

Specific Responses

Reviewer #1 (Remarks to the Author):

1. This paper by Sanchez-Murillo and colleagues presents a really valuable dataset on precipitation, surface water, and groundwater isotope ratios from Costa Rica. In addition to a long-term record, the authors conducted intensive sampling to characterize precipitation isotopes during the passage of a hurricane. This is of potential importance given the interest in using isotopes to trace the impact of severe storms on water resources and in reconstructing past storm activity using isotopic proxy data.

Reply 1: We appreciate all the valuable comments, suggestions and merit on data collection provided by Reviewer #1.

2. The records are unique, but I find the analysis and interpretation in the manuscript, as submitted, lacking. On the one hand, the authors present a number of interpretations, many emphasized as major conclusions (e.g., convective vs stratiform precipitation as a control on storm isotope ratios, estimates of uncertainty in paleo-proxy interpretations using the amount effect, and dismissal of moisture source as an isotopic control, all of which appear in the abstract), that are not at all supported by the analyses presented here. Conversely, I feel that they miss several important opportunities to analyze their data in ways that could really speak to the issue of paleoclimate reconstruction signals, emphasized in the title but given second-shrift in the paper. These issues are described in more detail below.

Reply 2: The reviewer rightfully pointed out that we omitted information regarding the relationship between the isotope ratios and the stratiform rainfall area fractions. This relevant issue was resolved by including a robust calculation of stratiform rainfall area fractions from available satellite data (GPM) for one of the longest isotope records of daily data on the Caribbean coast of Costa Rica. Full details about this new analysis can be found in the revised methods section. One of the reasons of submitting this revision after four months relies on the computational time needed to produce the analysis required by the reviewer.

As recommended by Reviewer 2 and due to the lack of a series of speleothem records from Costa Rica and the immediate surrounding area (nearest records are from Guatemala, Belize and Mexico), the focus of the manuscript was shifted towards an improved understanding of the dynamics and processes found in the observed isotope patterns. In this regard, we also applied spectral analysis to better characterize the inter-annual variability of the daily isotope records. Furthermore, the Vietnam-Costa Rica comparison to challenge the moisture source rationale was eliminated (and will be used to prepare a pan-tropical paper including other observational sites that exhibit the same pattern), as was the uncertainty estimate on paleo-rainfall reconstructions.

3. line 335: I find most of the discussion in this section to be poorly supported. My impression is that the authors are overlooking the basic first-order patterns in their data and instead rushing to an overly complex and inadequately justified interpretation framed in terms of a current 'hot' idea (stratiform/convective precipitation).

Reply 3: With all due respect, within our group, we are not simply rushing to frame our interpretations on the basis of a current 'hot idea'. Our results attempt to shed light on the classical 'amount effect' rationale used for decades without a sound physically-based

explanation. For instance, our observational results clearly show that enriched isotope ratios do not only occur during dry periods (meaning low rainfall regimes) and vice versa. Multiple interpretations based on paleo-proxy records simply concluded that enriched isotope ratios represent droughts and depleted ratios denote wet episodes, whereas modern rainfall isotope ratios are indicating a new potential explanation. Furthermore, these interpretations have been used to understand the collapse of major Mesoamerican civilizations, such as the Mayas, looking for past understanding to enhance our current climate change adaptation strategies. We do not find the stratiform and convective rationale overly complex nor an inadequate interpretation, since the mechanisms that cause isotope variability in rainfall are directly linked to precipitation forming systems and have been recently demonstrated across the globe:

- Aggarwal, P.K., Romatschke, U., Araguas-Araguas, L., Belachew, D., Longstaffe, F.J., Berg, P., Schumacher, C. and Funk, A., 2016. Proportions of convective and stratiform precipitation revealed in water isotope ratios. *Nature Geoscience*, 9(8), p.624.
- Lacour, J.L., Risi, C., Worden, J., Clerbaux, C. and Coheur, P.F., 2018. Importance of depth and intensity of convection on the isotopic composition of water vapor as seen from IASI and TES δD observations. *Earth and Planetary Science Letters*, 481, pp.387-394.
- Wei, Z., Lee, X., Liu, Z., Seeboonruang, U., Koike, M. and Yoshimura, K., 2018. Influences of large-scale convection and moisture source on monthly precipitation isotope ratios observed in Thailand, Southeast Asia. *Earth and Planetary Science Letters*, 488, pp.181-192.
- Aggarwal, P.K., Belachew, D., Schumacher, C., Funk, A.B., Longstaffe, F.J. and Terzer, S., 2016, December. What governs the oxygen and hydrogen isotopic composition of precipitation?-A case for varying proportions of isotopically-distinct, convective and stratiform rain fractions. In *AGU Fall Meeting Abstracts*.
- Zwart, C., Munksgaard, N.C., Protat, A., Kurita, N., Lambrinidis, D. and Bird, M.I., 2018. The isotopic signature of monsoon conditions, cloud modes, and rainfall type. *Hydrological Processes*, 32(15), pp.2296-2303.
- Lekshmy, P.R., Midhun, M. and Ramesh, R., 2018. Influence of stratiform clouds on δD and $\delta^{18}O$ of monsoon water vapour and rain at two tropical coastal stations. *Journal of Hydrology*.

4. To first order, the main pattern I see in the hurricane time series is one that has been shown in datasets from other such storms (e.g., Olivia, Sandy), which is high values corresponding to the outer rain bands and dropping values as 1) the storm's eye approaches and more 'processed' moisture from inner rain bands is delivered, and 2) the storm's circulation begins to interact more strongly with land and less with the ocean. This can largely be explained in terms of upwind rainout and Rayleigh theory (Good et al., 2014).

Reply 4: We agree with the reviewer on this comment. However, we have included data collected during hurricanes Irma and Maria (from the Bahamas and Cuba) in this new version to provide evidence of large TC systems producing fairly enriched isotope ratios with high rainfall volumes. It is also important to highlight that available TC isotope records correspond to the eastern and western Pacific Ocean or sub-tropical storms such as Ita or Sandy. However, our observations are the first high frequency and systematic measurements within the Mesoamerican and Caribbean region.

5. The roles of different moisture sources and stratiform/convective fractions are also worth considering, certainly, but the manuscript does not present any analyses that directly address these issues. I discuss concerns with the dismissal of vapor source controls below, and here just would point out that no data quantifying convective vs. stratiform precipitation are presented. This discussion section makes it sound as if the paper establishes these relations as truth, but no analyses are provided to back them up!

Reply 5: Idem Reply 2. As rightfully pointed out by the reviewer, information was lacking regarding the relationship between the isotope ratios and the stratiform rainfall area fractions (see comment above). This relevant issue was resolved by including a robust calculation of stratiform rainfall area fractions from available satellite data (GPM) for one of the longest isotope records of daily data on the Caribbean coast of Costa Rica. Full details about this new analysis can be found in the revised methods section and the time-consuming computation is also the reason for our delayed re-submission.

6. line 360-363: Not clear how increased humidity and wind speed produce negative d -excess values. I think you can get to zero this way but not below. What is the proposed mechanism here?

Reply 6: Several studies have reported negative d -excess values in both water vapor and precipitation and its anti-correlation with high relative humidity values (see more detailed explanations in general comments above).

- Klein ES, Cherry JE, Young J, Noone D, Leffler AJ, Welker JM. Arctic cyclone water vapor isotopes support past sea ice retreat recorded in Greenland ice. *Scientific reports*. 2015 May 29;5:10295.
- Aemisegger, F., Pfahl, S., Sodemann, H., Lehner, I., Seneviratne, S.I. and Wernli, H., 2014. Deuterium excess as a proxy for continental moisture recycling and plant transpiration. *Atmospheric Chemistry & Physics*, 14(8).
- Wang, S., Zhang, M., Che, Y., Zhu, X. and Liu, X., 2016. Influence of below-cloud evaporation on deuterium excess in precipitation of arid central Asia and its meteorological controls. *Journal of Hydrometeorology*, 17(7), pp.1973-1984.
- Cui, J., Tian, L., Biggs, T.W. and Wen, R., 2017. Deuterium-excess determination of evaporation to inflow ratios of an alpine lake: Implications for water balance and modeling. *Hydrological processes*, 31(5), pp.1034-1046.
- Sánchez-Murillo, R., Durán-Quesada, A.M., Birkel, C., Esquivel-Hernández, G. and Boll, J., 2017. Tropical precipitation anomalies and d -excess evolution during El Niño 2014-16. *Hydrological Processes*, 31(4), pp.956-967.

7. l 414: What is the basis for the dismissal of moisture source as a control on isotope ratios here? The referenced figure (6) provides no such analysis, and the text provides no justification. The subsequent reference to a comparison between Costa Rica and Vietnam seasonal patterns is trivial – although clearly these sites in different parts of the world receive moisture from different parts of the ocean there could be a common seasonal pattern in moisture source conditions (and isotope ratios). I'm not saying that I think this is true, just pointing out that the manuscript provides no concrete evidence against it. You would need to conduct moisture source attribution analysis to actually test the idea.

Reply 7: This comparison was eliminated from the revised manuscript based on the suggestions by both reviewers.

8. Figure 6: The comparison shown in panel B is very interesting...are the statistical distributions of Otto-associated d18O and the rest of the record at all different? It sure doesn't look like it. Nonetheless the surface and groundwater data show a spike, so I'm guessing that the amount-weighted rainfall for the storm looks very different than the unweighted mean of these fixed interval samples (why not present this comparison?). One could ask the question 'what would it take to actually detect this event (Otto) in a paleo-record', which is a very direct question that is at the heart of this manuscript but never actually addressed. The authors have an outstanding dataset to work with (both the precip and the sw and gw) and I would love to see them design some numerical experiments to really test this idea (e.g., simulate attenuation in different proxies types and ask whether an Otto signal stands out above the noise). IMHO this would be a more significant contribution than the tact the manuscript currently takes.

Reply 8: We appreciate the Reviewer's thoughtful and helpful suggestions related to Figure 6. However, those statistical distributions were not different, since Otto produced the entire isotope range observed within a regular water year in Costa Rica. The main idea of presenting surface and groundwater isotope compositions during the passage of the storm is to provide evidence of the rapid pulse of the storm through different water reservoirs. We now have re-directed the manuscript towards the understanding of the dynamics and processes that lead to the observed isotope patterns and for this reason no proxy simulations were conducted at this point.

9. line 486-488: Is this true given the high degree of time-averaging (attenuation) of the paleo archive signals? I don't think so.

Reply 9: This sentence was eliminated.

10. line 488-492: As mentioned above, this has not been supported

Reply 10: This sentence was eliminated.

A reference was already included in the text which is also related to the suggested reference below, therefore we do not find appropriate to include a similar citation.

Good, S. P., Mallia, D. V., Denis, E. H., Freeman, K. H., Feng, X., Li, S., . . . Bowen, G. J. (2014). High frequency trends in the isotopic composition of Superstorm Sandy. In J. B. Bennington & E. C. Farmer (Eds.), Learning from the Impacts of Superstorm Sandy (pp. 41-55): Academic Press.

Reviewer #2 (Remarks to the Author):

1. "Hurricane isotope anatomy in a climate change sensitive region: implications for tropical paleoclimatic re-constructions" presents a comprehensive new water isotope dataset from Costa Rica of daily precipitation sampled at multiple sites across the country as well as spring and surface waters. The dataset has captured a rare, southern hurricane path, and the authors combine precipitation and isotope data to describe the expression of the storm in isotope-space, and to examine the assumption that the tropical 'amount effect' holds on daily/event timescales, and what this all means in terms of interpreting isotope-based proxy data, commonly used to reconstruct past tropical cyclones and hurricanes. The dataset is really fantastic and figures are also quite attractive and high quality. The authors have put a lot of work into this ms.

Reply 1: We highly appreciate all the valuable comments, suggestions and merit on data collection provided by Reviewer #2.

2. The main take home message I am getting is, as in many sites and recent papers, the classical tropical 'amount effect' does not hold on the event timescale in Costa Rica, generally, and when considering this hurricane. This is fine with me, but I think the authors could go further with their analysis of this dataset. For example, In the second to last paragraph, the authors conclude that what is driving isotopic variability is "convective activity and ... fractions of shallow convective and stratiform rainfall generated, suggesting a major role of the in-cloud microphysics and thermodynamics (i.e., latent heat profile) within the precipitable water column, particularly between the convective and stratiform region in Mesoscale systems. Yet, there are no figures or analysis of % stratiform vs convective precip vs isotope values, OLR, or other meteorologic variables vs precip isotope values, so I don't see how this conclusion is drawn.

Reply 2: Absolutely and based on these comments we now included new analysis on the relationship between the isotope ratios and the stratiform rainfall area fractions. Please, see our response to Reviewer 1 and in the general comments above for more details.

3. My biggest issue with the manuscript as it is written is the framework in which the data are presented and analyzed. The science problem the authors pose in the introduction is testing the 'amount effect' (and the moisture source effect, although they don't expand upon this as much) in order to properly interpret isotope-based paleoclimate proxies. However, the article fails to come 'full circle' and bring the reader back to any specific, or radical re-interpretation of proxy records. It's also unclear how many proxy records there are, and how many need to be reinterpreted etc., and what the implications of those re-interpretations may be. I suggest either 1) digging deeply into the paleodata or proxy system modeling to show how your results have clear impacts on interpretations, or 2) reframing the paper to focus more on understanding the dynamics and processes that lead to the observed isotope patterns. In that light, when introducing the amount effect, it is important to acknowledge the multiple hypotheses (many supported by observations) proposed in papers over the last decade regarding what produces the amount effect. The authors have a great vehicle in this dataset in which to systematically test some of these hypotheses.

Reply 3: As recommended by Reviewer 2 and due to the lack of a series of speleothem records, we shifted the focus of the revised paper towards an improved understanding of the dynamics and processes that lead to the observed isotope patterns. In this regard, the revised manuscript now includes:

- a) Spectral analysis of $\delta^{18}\text{O}$ isotope variability**
- b) Calculation of stratiform rainfall area fractions from satellite data**
- c) Addition of new rainfall isotope data from the major hurricanes Irma and Mari**
- d) Reframe the paper to focus more on understanding the dynamics and processes that lead to the observed isotope patterns**
- e) Deletion of the stable isotope comparison between Costa Rica and Vietnam, since this component was slightly out of the scope of the manuscript**

4. How many isotope based paleoclimate records from your region actually convert proxy $\text{d}18\text{O}$ in mm/day space? The authors discuss this transfer function, but I wonder how often $\text{d}18\text{O}$ is really converted.

Reply 4: Rainfall re-constructions are usually done on annual or decadal basis. Attached are some examples.

- Tozer, C.R., Vance, T.R., Roberts, J.L., Kiem, A.S., Curran, M.A. and Moy, A.D., 2016. An ice core derived 1013-year catchment-scale annual rainfall reconstruction in subtropical eastern Australia. *Hydrology and Earth System Sciences*, 20(5), pp.1703-1717.
- Lachniet, M.S., Asmerom, Y., Polyak, V. and Bernal, J.P., 2017. Two millennia of Mesoamerican monsoon variability driven by Pacific and Atlantic synergistic forcing. *Quaternary Science Reviews*, 155, pp.100-113.
- Pollock, A.L., 2015. Mid-Holocene Speleothem Climate Proxy Records from Florida and Belize.

5. I don't see the value in comparing this record from Costa Rica to a record from Vietnam, it does not add much to the manuscript. The conclusion you are trying to make (about moisture source impact on precip d18O?) is not coming across.

Reply 5: This comparison was eliminated as recommended by both reviewers.

6. In terms of the authors concluding that moisture source is not important, I am struck by panel A in figure 2, which shows 2 disturbances, one in the Pacific, and one in the Caribbean. What about teasing out Pacific vs Caribbean moisture contributions to this event with some mixing models? And there is brief mention of a Nicaraguan lake moisture source contribution, but this is not analyzed further?

Reply 6: The main water-energy source of this event was the southern Caribbean Sea. However, at the beginning of the Tropical Storm system, two convective nuclei were located in the south Pacific and Caribbean Sea. Later, as the storm turned into a hurricane, the main vapor source came from the Caribbean Sea. The reference of the Nicaraguan lake relies on the unusual crossover of the storm as a hurricane from the Caribbean Sea to the Pacific Ocean, which most likely was strengthened by the relative warm waters of the lake. This is now further explained in the revised paper.

7. A major conclusion is the imprint of a single extreme event on surface and subsurface water (SF2). That is impressive and somewhat glossed over. These data are only talked about in terms of damping, but what about considering lag times, and the magnitude of isotope depletion required to be imprinted on surface and subsurface water? Building a pseudo-speleothem with a proxy system model (these are out there and not too complicated)? This is the information that directly translates to speleothem records.

Reply 7: Caribbean-type rainfall is often enriched with mean values ranging from ~1‰ to -5‰, with nearly constant rainfall throughout the year, and water recharge corresponds to rather short travel times. Isotope composition in one of the springs shows the TC signal with a minimal lag time. We now provided more explanation in the revised paper.

8. The discussion of the range of isotope values in this storm suggests that a pdf type analysis, looking at tails (skewness) could also be really useful.

Reply 8: This section was significantly improved and new data collected during hurricanes Irma and Maria are now included.

Reviewers' comments:

Reviewer #1 (Remarks to the Author):

My opinion of this manuscript remains largely unchanged from the previous version. I think it is a unique and valuable dataset with some interesting features and relevant to some important issues. However, I think that the manuscript still suffers from incomplete or imprecise reasoning in places and I am reluctant to support its publication as a result. Part of this is presentation, part is depth of analysis. Accompanying this, the manuscript presents at least 3 pretty distinct datasets and I think struggles to integrate them in a coherent way. I could see a strong manuscript that looked at atmospheric processes across scales and systems (combining all the precip data) or one that dealt w/ cyclone signatures in meteoric waters (Otto, springs and surface water). Perhaps partially because this manuscript tries to address both topics I don't feel that either gets a really robust treatment here and the text is less coherent than it could be.

The primary contention in the revised manuscript is that the amount effect is not a good description for isotope ratios in the study area, and that instead the ratio of stratiform to convective rainfall controls these values. The first part of this is supported by the data, at least in the context of the classical, locally-defined amount effect (correlation between local rainfall amounts and isotope ratios, here weak but significant). Some paleoclimate work continues to reference the amount effect, which makes this a useful observation, though I'd note that I don't know anyone who truly is thinking about isotope systematics who argues that the classical amount effect is an adequate description of the phenomena controlling tropical rainfall isotope values. The second part was not clearly supported by data or analyses in the original submission, nor is it here.

In fact, the authors have added a nice new analysis that shows that the daily isotope data are strongly correlated with precipitation amount; not at the local level, but regionally (figure 4B, where the x-axis data are regional precipitation amounts, transformed by the authors as a proxy for stratiform area). This has been observed before, for example by (Kurita et al., 2009).

I seem to have offended with my previous comment about 'hot ideas', and I apologize if so. I was not questioning the authors' scientific integrity, or any specific idea, but rather pointing out that the manuscript bypassed simple explanations for the data and instead advocated for more complex ones without any robust evidence requiring this. Here we have a nice example. Classical Rayleigh theory would say that if regional precipitation rates are higher, and some of the vapor from that region is being advected to the site where the authors are sampling, that this would result in lower precipitation isotope values at the sample site. This is the simplest and most direct mechanism consistent with what their new analysis is showing. Does explaining the results require more than this? No case is made.

The same figure also shows that when regional precipitation rates are higher, this is usually correlated with widespread stratiform rainfall in the region (Fig. 4A). This begs the question 'does it matter that the rain-producing systems are stratiform?'. 'If the same amount of rain fell and it was all via convection would the isotope values be different?' I certainly believe it's possible. There are good first principles reasons to think that the atmospheric water balance differs with storm type, and this has been discussed and direct observations offered in some of the papers referenced. But there are no data provided here to test this idea.

So, what do these data, and presumably paleo-isotope archive records from the region, tell us about? Regional precipitation intensity? I'm on board w/ that based on the analyses here. Regional stratiform rainfall intensity? OK, they show that this is how the modern climate works, though with the caveat that strong, well-organized mesocyclones (e.g., Otto) dominated by convection can locally produce very light precipitation. Local or regional stratiform:convective rainfall fractions? A possible influence, but not something that this manuscript tests or demonstrates. If this is how they talked about their findings I'd be on board. Late in the text they suggest that paleo-records might indicate changes in the amount of deep convection, presumably implying that more deep convection would give higher isotope values. My personal reaction is that this makes no sense, but regardless it's not something that follows from any analysis presented here.

I've highlighted some places that illustrate the general issue below. Just to emphasize, I've used the stratiform:convective topic as an example but there are other topics discussed where there's a similar imprecision in the way results are presented.

Line numbers refer to the version w/o edits shown.

L272: What is 3B42?

L287-390: Not to defend the amount effect, but this is essentially what you're describing here. At seasonal timescales your data vary inversely w/ patterns of precipitation amount.

L309-311: This statement is not justified by the figure or analysis. You have no analysis that shows that strong convective cells result in Costa Rica precipitation values of +4 to -6. The referenced figure and analysis convey no information about convection, period.

L321-324: These statements are speculation but presented as if they are fact. Yes, these features exist, but you have not shown that they cause the isotopic patterns.

L330: known based on what? Do you mean expected? Include a reference to justify the expectation.

L350-353: Here again careful with the language. You are reporting correlations, you have no analysis that allows definitive attribution of the d-excess values to a particular mechanism. The wording in the first half of the sentence is probably OK, that in the second half is overstating the result.

L368: Here the word unique is puzzling to me. You haven't really shown to this point that the storm water is, on the whole, unique in its isotope ratios. Actually most of what you've presented seems to emphasize that most of the cyclone water is pretty 'normal'. Can you clarify what you mean here...I think you're talking about the small number of very ^{18}O -depleted samples associated with storm core precipitation, right?

L374+: Is discussing the attenuation in terms of per mil very informative? I'd think it would be more natural to disuse the amplitude attenuation in relative terms (input-background/signal-background) to allow for the fact that different storms will have different isotopic extreme values. You could then talk about this in terms of delta as it relates to the identifiability of the signal, e.g., how big a precipitation isotope anomaly would you need to see it.

L397-401: The noise around the correlation you report shows that factors other than local precipitation amount control most of the isotope variation in this system. It allows for the possibility that the stuff you describe in this sentence is some or all of what controls the isotope variation, but doesn't provide any direct support (I'm not sure what "reinforces" means in this context, but I read it to be more than "is consistent with the possibility that...", which is what your data show). The result referenced here (+ all of your analyses) say nothing about the role of moisture source, that I can see, so I don't think it's necessary or appropriate to mention this.

L415-416: This is inaccurate. None of the analyses here show a direct link between 'changes in precipitation types' and isotopes. The closest might be the timeseries evolution of Otto, but the analysis presented there was purely descriptive. None of your analyses quantify 'convective activity'. The rest of the paragraph builds from this and, while written more as speculation and so technically probably OK, comes across as misleading.

L433-438: Same issue...

Kurita, N., Ichiyangi, K., Matsumoto, J., Yamanaka, M.D., Ohata, T., 2009. The relationship between the isotopic content of precipitation and the precipitation amount in tropical regions. *Journal of Geochemical Exploration*, 102: 113-122.

Reviewer #2 (Remarks to the Author):

This revised version of Sanchez-Murillo et al. is much improved and I recommend it be accepted. My comments are minimal—

Much emphasis is placed on the 'new framework' for interpreting tropical precipitation isotope values, but the actual framework is not completely clear to me, apart from the concept of changing isotope ratios as the storm progresses, independent of precipitation amount. Readers would benefit from a simplified cartoon schematic of said framework.

The scatter plots of $\delta^{18}O$ vs stratiform rain and precip indicate non-normally distributed data. For consideration of the strength of the relationship between 2 variables then, a nonparametric test is required, like Spearman's rank correlation coefficient (Martin et al., 2018 JGR-A)

Some mention of temporal scale is warranted in the manuscript. The authors are focused on the event scale, where processes within storms determine isotopic values. But most paleoclimate and previous isotope interpretations of the last several decades are based on monthly or longer timescales, and consider controls on isotope values from more of an integrated, or climatological perspective.

What happens when you consider the same relationships in your dataset when the data are in monthly average form? Do the relationships with stratiform rain or precipitation amount change?

Reviewer #1 (Remarks to the Author):

My opinion of this manuscript remains largely unchanged from the previous version. I think it is a unique and valuable dataset with some interesting features and relevant to some important issues. However, I think that the manuscript still suffers from incomplete or imprecise reasoning in places and I am reluctant to support its publication as a result. Part of this is presentation, part is depth of analysis. Accompanying this, the manuscript presents at least 3 pretty distinct datasets and I think struggles to integrate them in a coherent way. I could see a strong manuscript that looked at atmospheric processes across scales and systems (combining all the precip data) or one that dealt w/ cyclone signatures in meteoric waters (Otto, springs and surface water). Perhaps partially because this manuscript tries to address both topics I don't feel that either gets a really robust treatment here and the text is less coherent than it could be.

R/ Thanks again for providing great comments. In order to ensure a clear message, we decided to eliminate the surface and groundwater aspect of the manuscript. Therefore, the manuscript only deals now with atmospheric processes and rainfall isotope variations including tropical cyclone observations.

The primary contention in the revised manuscript is that the amount effect is not a good description for isotope ratios in the study area, and that instead the ratio of stratiform to convective rainfall controls these values. The first part of this is supported by the data, at least in the context of the classical, locally-defined amount effect (correlation between local rainfall amounts and isotope ratios, here weak but significant). Some paleoclimate work continues to reference the amount effect, which makes this a useful observation, though I'd note that I don't know anyone who truly is thinking about isotope systematics who argues that the classical amount effect is an adequate description of the phenomena controlling tropical rainfall isotope values. The second part was not clearly supported by data or analyses in the original submission, nor is it here.

R/ The amount effect discussion is now presented from a traditional and statistical perspective, all comments not well supported by clear evidence were removed.

In fact, the authors have added a nice new analysis that shows that the daily isotope data are strongly correlated with precipitation amount; not at the local level, but regionally (figure 4B, where the x-axis data are regional precipitation amounts, transformed by the authors as a proxy for stratiform area). This has been observed before, for example by (Kurita et al., 2009). I seem to have offended with my previous comment about 'hot ideas', and I apologize if so. I was not questioning the authors' scientific integrity, or any specific idea, but rather pointing out that the manuscript bypassed simple explanations for the data and instead advocated for more complex ones without any robust evidence requiring this. Here we have a nice example.

Classical Rayleigh theory would say that if regional precipitation rates are higher, and some of the vapor from that region is being advected to the site where the authors are sampling, that this would result in lower precipitation isotope values at the sample site. This is the simplest and most direct mechanism consistent with what their new analysis is showing. Does explaining the results require more than this? No case is made.

R/ The manuscript is now presented with clear and simple explanations.

The same figure also shows that when regional precipitation rates are higher, this is usually correlated with widespread stratiform rainfall in the region (Fig. 4A). This begs the question 'does it matter that the rain-producing systems are stratiform? 'If the same amount of rain fell and it was all via convection would the isotope values be different?' I certainly believe it's possible. There are good first principles reasons to think that the atmospheric water balance differs with storm type, and this has been discussed and direct observations offered in some of the papers referenced. But there are no data provided here to test this idea. So, what do these data, and presumably paleo-isotope archive records from the region, tell us about? Regional precipitation intensity? I'm on board w/ that based on the analyses here. Regional stratiform rainfall intensity? OK, they show that this is how the modern climate works, though with the caveat that strong, well-organized mesocyclones (e.g., Otto) dominated by convection can locally produce very light precipitation. Local or regional stratiform:convective rainfall fractions? A possible influence, but not something that this manuscript tests or demonstrates. If this is how they talked about their findings I'd be on board. Late in the text they suggest that paleo-records might indicate changes in the amount of deep convection, presumably implying that more deep convection would give higher isotope values. My personal reaction is that this makes no sense, but regardless it's not something that follows from any analysis presented here. I've highlighted some places that illustrate the general issue below. Just to emphasize, I've used the stratiform:convective topic as an example but there are other topics discussed where there's a similar imprecision in the way results are presented.

R/ The manuscript discusses now only arguments supported by our results, speculative arguments were removed.

Line numbers refer to the version w/o edits shown.

L272: What is 3B42?

R/ Typo removed.

L287-390: Not to defend the amount effect, but this is essentially what you're describing here. At seasonal timescales your data vary inversely w/ patterns of precipitation amount.

R/ The reviewer is obviating, for example, that during the strongest trade winds rainfall amounts are large but the composition is not depleted at all. The amount effect is not applicable year around, and therefore data cannot be considered as a simple inverse function of precipitation patterns as suggested.

L309-311: This statement is not justified by the figure or analysis. You have no analysis that shows that strong convective cells result in Costa Rica precipitation values of +4 to -6. The referenced figure and analysis convey no information about convection, period.

R/ Statement was rephrased, and a set of published papers was included to support this argument. As basic as it is, rainfall is fundamentally of two types across the globe: convective and stratiform.

L321-324: These statements are speculation but presented as if they are fact. Yes, these features exist, but you have not shown that they cause the isotopic patterns.

R/ Statement was removed.

L330: known based on what? Do you mean expected? Include a reference to justify the expectation.

R/ Statement was rephrased.

L350-353: Here again careful with the language. You are reporting correlations, you have no analysis that allows definitive attribution of the d-excess values to a particular mechanism. The wording in the first half of the sentence is probably OK, that in the second half is overstating the result.

R/ Statement was rephrased.

L368: Here the word unique is puzzling to me. You haven't really shown to this point that the storm water is, on the whole, unique in its isotope ratios. Actually most of what you've presented seems to emphasize that most of the cyclone water is pretty 'normal'. Can you clarify what you mean here...I think you're talking about the small number of very ^{18}O -depleted samples associated with storm core precipitation, right?

R/ Section was eliminated.

L374+: Is discussing the attenuation in terms of per mil very informative? I'd think it would be more natural to disuse the amplitude attenuation in relative terms (input-background/signal-background) to allow for the fact that different storms will have

different isotopic extreme values. You could then talk about this in terms of delta as it relates to the identifiability of the signal, e.g., how big a precipitation isotope anomaly would you need to see it.

R/ Section was eliminated.

L397-401: The noise around the correlation you report shows that factors other than local precipitation amount control most of the isotope variation in this system. It allows for the possibility that the stuff you describe in this sentence is some or all of what controls the isotope variation, but doesn't provide any direct support (I'm not sure what "reinforces" means in this context, but I read it to be more than "is consistent with the possibility that...", which is what your data show). The result referenced here (+ all of your analyses) say nothing about the role of moisture source, that I can see, so I don't think it's necessary or appropriate to mention this.

R/ Section was modified.

L415-416: This is inaccurate. None of the analyses here show a direct link between 'changes in precipitation types' and isotopes. The closest might be the timeseries evolution of Otto, but the analysis presented there was purely descriptive. None of your analyses quantify 'convective activity'. The rest of the paragraph builds from this and, while written more as speculation and so technically probably OK, comes across as misleading.

R/ Agree, section was modified.

L433-438: Same issue...

R/ Agree, section was modified.

Kurita, N., Ichiyanagi, K., Matsumoto, J., Yamanaka, M.D., Ohata, T., 2009. The relationship between the isotopic content of precipitation and the precipitation amount in tropical regions. *Journal of Geochemical Exploration*, 102: 113-122.

Reviewer #2 (Remarks to the Author):

This revised version of Sanchez-Murillo et al. is much improved, and I recommend it be accepted. My comments are minimal.

R/ Thanks for your revision and great comments.

Much emphasis is placed on the 'new framework' for interpreting tropical precipitation isotope values, but the actual framework is not completely clear to me, apart from the concept of changing isotope ratios as the storm progresses, independent of

precipitation amount. Readers would benefit from simplified cartoon schematic of said framework.

R/ Thanks for providing this great perspective, we have modified the framework and data sets are now combined and analyzed in more coherent manner to deliver a clear scientific message.

The scatter plots of d18O vs stratiform rain and precip indicate non-normally distributed data. For consideration of the strength of the relationship between 2 variables then, a nonparametric test is required, like Spearman's rank correlation coefficient (Martin et al., 2018 JGR-A). Some mention of temporal scale is warranted in the manuscript. The authors are focused on the event scale, where processes within storms determine isotopic values. But most paleoclimate and previous isotope interpretations of the last several decades are based on monthly or longer timescales and consider controls on isotope values from more of an integrated, or climatological perspective. What happens when you consider the same relationships in your dataset when the data are in monthly average form? Do the relationships with stratiform rain or precipitation amount change?

R/ A new statistical analysis comparing amount effects at different timescales is provided.

Reviewers' comments:

Reviewer #2 (Remarks to the Author):

This manuscript is much improved and should be published. There are a few typos here and there, which should be corrected:

L40: remove s from TCs

L46: remove s from TCs

L47: whose is used for people, not ideas or things, reword

L73: delete 'to'

L92: change isotopic composition to d18O, also define d18O when you use it the first time

L100: composition cannot be pluralized with s. Replace with values

L139: delete of

L181: Lake Nicaragua is the formal name, not Nicaraguan Lake

(This statement is very interesting, btw)

L205: delete 'information of the', to improve sentence

L297: replace frequency with periodicity to match your figure.

L305: Add 'a' between such and mechanism

Conclusion: reiterate L337-340. This is a powerful observation that should not get lost in the main text alone.

Figure captions: Make sure Hurricane Otto is consistently capitalized.

I have also been asked to comment on the response to reviewer #1's comments. I think the authors have provided satisfying responses and associated corrections. I agree that it is a stronger paper without the surface water component.

Nice work!

Reviewer #3 (Remarks to the Author):

This manuscript presents new high-resolution precipitation isotope data collected over a period of 5 years in Costa Rica, as well as at sites across the Mesoamerican and Caribbean region during the passage of Hurricanes Otto in 2016 and Hurricanes Irma and Maria in 2017. These valuable datasets are useful for investigating isotopic signatures associated with Tropical Cyclones (TCs), which is important for efforts to use oxygen isotope-based proxy records to reconstruct past extreme precipitation events. The authors also utilize these to evaluate the influence of rainfall amount (the "amount effect") and cloud type on precipitation isotopes in the region. They basically conclude that there is substantial variability in precipitation isotopes during the TCs (both in d18O and d-excess), with the more maritime Irma and Maria storms characterized by relatively positive d18O values. Otto, on the other hand showed initially high d18O followed by rapid depletion as the storm made landfall, similar to previously reported strong isotopic depletions observed during continental TC tracks. They also note d-excess decreases which they link to "highly efficient isotopic fractionation" – though the paper doesn't go in great depth in explaining this. They further hypothesize that stratiform vs convective activity may be more important than rainfall amount in driving precipitation isotope variability in the region.

A key motivation of the study, according to the authors is to "inform paleo-water isotope interpretations related to key climate drivers in tropical TC activity", but on this point, I feel this study falls short. In particular, I agree with the previous Reviewer 1, that in the end – the data presented here supports a general "amount effect" based interpretation of precipitation isotope proxy records. For instance, in Figure 7B, the authors show that 28-88% of the variance can be explained by the amount effect on monthly timescales (though in many cases, it is better to look at interannual slopes, though this is rarely possible). When comparing this with Figure 4B, for which no statistics are given, it does not obviously look like the stratiform rainfall area – d18O correlation is any stronger than the monthly rainfall "amount effect" correlation – calling in to question why the authors chose to focus on this idea. Similarly, Figure 4A also shows a strong

correlation between stratiform rain area and regional precipitation, lending further support to the amount effect. I also agree, with the earlier review, that few studies base their proxy interpretation solely on the "classical amount effect", but rather on a more nuanced analysis that takes in to account other factors such as moisture source region, etc. Also, relevant is the fact that proxy archives integrate over many months to years, so the type of high-frequency regional variability discussed here is less relevant when discussing the paleoclimate record. Along these same lines, few studies attempt to use a quantitative transfer function to reconstruct precipitation amount, as the authors imply, so the suggestion that these proxies may only be interpreted qualitatively is pretty much in line with the status quo. I would therefore argue that the better use of this dataset may be to inform on the dynamics of the storms themselves, but on this point, there is also not much new added by this manuscript.

Overall – while the dataset is clearly valuable and will be of great interest to the tropical paleoclimate, water isotope, and TC communities, I do not think the findings are novel enough or will be of broad enough interest to warrant publication in Nature Communications.

I also have several specific comments/edits, as follows:

Lines 40 and 46 – "TCs" does not need the "s" here and elsewhere when not plural.

Line 71 – It is not clear what "latter" is referring to

Line 73 – delete "to"

Lines 81-82 – worth also mentioning isotope enabled models that can also be used to investigate water isotope variability in data poor regions

Lines 87-90 – some references should be included here

Line 92 – "monthly" can be deleted – the amount effect does not specifically refer to monthly values

Line 98 – quantitative transfer functions are rarely used – though if they are, this is a valid critique

Lines 105 and 111 – replace "featured" with "characterized"

Detailed point by point revision No4

Reviewers' comments:

Reviewer #2 (Remarks to the Author):

This manuscript is much improved and should be published: **R/Thanks for your evaluation and helping us to improve our manuscript.**

There are a few typos here and there, which should be corrected:

L40: remove s from TCs: **R/Suggested edit completed.**

L46: remove s from TCs: **R/Suggested edit completed.**

L47: whose is used for people, not ideas or things, reword: **R/Suggested edit completed.**

L73: delete 'to': **R/Suggested edit completed.**

L92: change isotopic composition to d18O, also define d18O when you use it the first time: **R/Suggested edit completed.**

L100: composition cannot be pluralized with s. Replace with values: **R/Suggested edit completed.**

L139: delete of: **R/Suggested edit completed.**

L181: Lake Nicaragua is the formal name, not Nicaraguan Lake. (This statement is very interesting, btw): **R/ Name changed.**

L205: delete 'information of the', to improve sentence: **R/Suggested edit completed.**

L297: replace frequency with periodicity to match your figure: **R/Suggested edit completed.**

L305: Add 'a' between such and mechanism: **R/Suggested edit completed.**

Conclusion: reiterate L337-340. This is a powerful observation that should not get lost in the main text alone: **R/Suggested edit completed.**

Figure captions: Make sure Hurricane Otto is consistently capitalized: **R/Suggested edit completed, Hurricane Otto is consistently capitalized**

I have also been asked to comment on the response to reviewer #1's comments. I think the authors have provided satisfying responses and associated corrections. I agree that it is a stronger paper without the surface water component.

R/Thanks again for your thoughtful and positive feedback during this revision. This is now noted in the Acknowledgments.

Reviewer #3 (Remarks to the Author):

This manuscript presents new high-resolution precipitation isotope data collected over a period of 5 years in Costa Rica, as well as at sites across the Mesoamerican and Caribbean region during the passage of Hurricanes Otto in 2016 and Hurricanes Irma and Maria in 2017. These valuable datasets are useful for investigating isotopic signatures associated with Tropical Cyclones (TCs), which is important for efforts to use oxygen isotope-based proxy records to reconstruct past extreme precipitation events.

We thank the reviewer #3 for recognizing that our data sets are valuable and important. We would like to mention that in our revision, we have tried to emphasize that these datasets are to the best of our knowledge the first ground-based isotopic measurements in the Intra Americas Seas during TC passages. We are aware of only one previous effort that collected rainfall during aircraft missions 25 years ago (in 1994-1995) (Gedzelman et al., 2003). Therefore, our data represent the only complete, modern-day dataset about TC isotopic signatures available to evaluate past climate reconstructions in this key region.

Gedzelman, S. *et al.* Probing hurricanes with stable isotopes of rain and water vapor. *Monthly Weather Review* **131**, 1112-1127 (2003).

The authors also utilize these to evaluate the influence of rainfall amount (the “amount effect”) and cloud type on precipitation isotopes in the region. They basically conclude that there is substantial variability in precipitation isotopes during the TCs (both in d18O and d-excess), with the more maritime Irma and Maria storms characterized by relatively positive d18O values. Otto, on the other hand showed initially high d18O followed by rapid depletion as the storm made landfall, similar to previously reported strong isotopic depletions observed during continental TC tracks.

We agree with this summary statement. In addition, however, we have highlighted in our revision that previous studies were focused on extra-tropical cyclones or

typhons (such as e.g. Hurricane Sandy). It is rainfall type and not cloud type that we make reference to in the manuscript, and this was clarified in the text to avoid confusion. One critical difference between Otto's $\delta^{18}\text{O}$ signatures and those of the more maritime events (Irma and Maria) is the depletion caused by Otto's passage over the Central American Isthmus. While some previous authors suggest that the entire region can be viewed as a maritime regime when events develop in the southernmost portion of the MAC, we demonstrated that TC passage on a southern track is characterized by strong depletion similar to continental TC passages.

They also note d-excess decreases which they link to "highly efficient isotopic fractionation" – though the paper doesn't go in great depth in explaining this. They further hypothesize that stratiform vs convective activity may be more important than rainfall amount in driving precipitation isotope variability in the region.

We now provide further comment on d-excess in the manuscript as follows: "Deuterium excess changes are strongly related to relative humidity gradients. Large relative humidity gradients often result in large d-excess values, whereas high relative humidity values result in low d-excess values. As Hurricane Otto made landfall, efficient moisture transport increased relative humidity values, driving a reduction in d-excess values."

Regarding stratiform versus convective activity as a main driver for precipitation isotope signatures, we note that the d-excess changes coincide with the largest variations in precipitable water vapor (PWV). As the cloud bands migrate towards increasing PWV, convective rainfall activates, while rapid reductions in PWV correspond with stratiform rainfall as observed in the decaying cloud top shields in the satellite imagery, giving rise to more depleted values. The manuscript now includes a discussion in terms of the satellite rain band features and the implications for rainfall isotopic composition.

A key motivation of the study, according to the authors is to "inform paleo-water isotope interpretations related to key climate drivers in tropical TC activity", but on this point, I feel this study falls short.

We feel confident that our data will find widespread use in the paleo-hydrology community – given ongoing efforts to reconstruct TC's in a variety of isotopic archives. Please see the response to the next comment.

In particular, I agree with the previous Reviewer 1, that in the end – the data presented here supports a general "amount effect" based interpretation of precipitation isotope proxy records. For instance, in Figure 7B, the authors show that 28-88% of the variance can be explained by the amount effect on monthly timescales (though in many cases, it

is better to look at interannual slopes, though this is rarely possible). When comparing this with Figure 4B, for which no statistics are given, it does not obviously look like the stratiform rainfall area – $\delta^{18}\text{O}$ correlation is any stronger than the monthly rainfall “amount effect” correlation – calling in to question why the authors chose to focus on this idea. Similarly, Figure 4A also shows a strong correlation between stratiform rain area and regional precipitation, lending further support to the amount effect. I also agree, with the earlier review, that few studies base their proxy interpretation solely on the “classical amount effect”, but rather on a more nuanced analysis that takes in to account other factors such as moisture source region, etc. Also, relevant is the fact that proxy archives integrate over many months to years, so the type of high-frequency regional variability discussed here is less relevant when discussing the paleoclimate record. Along these same lines, few studies attempt to use a quantitative transfer function to reconstruct precipitation amount, as the authors imply, so the suggestion that these proxies may only be interpreted qualitatively is pretty much in line with the status quo. I would therefore argue that the better use of this dataset may be to inform on the dynamics of the storms themselves, but on this point, there is also not much new added by this manuscript.

R/ We thank the reviewer once again for a thorough discussion on this topic and would like to add that the storm dynamics were part of the author’s interest aiming to provide a detailed documentation of a rare event (the first) from an isotope perspective. Nonetheless, our interest and motivation in this work goes beyond documenting the important event with unique data. We are trying to encourage the community to consider alternative interpretations beyond the classical amount effect, while acknowledging that it does explain a portion of the variance observed in our dataset, and many others. The key point is that the specific processes in question, and their spatial-temporal footprints, will vary appreciably by region. For sites that are influenced by TC activity, our study would help to constrain the contribution of such events to lower-resolution records, keeping in mind that while they are short-lived, they could play an outsized role in amount-weighted rainfall $\delta^{18}\text{O}$ at a given site, given the large amounts of rainfall delivered during these events. Therefore, we urge the paleoclimate community to consider the full range of climatic phenomenon that contribute to rainfall $\delta^{18}\text{O}$ variability at their site, and in this sense our manuscript

provides quantitative constraints on a weather extreme that plays a key role in climate across Mesoamerica.

Statistics for Figure 4B are included now.

Overall – while the dataset is clearly valuable and will be of great interest to the tropical paleoclimate, water isotope, and TC communities, I do not think the findings are novel enough or will be of broad enough interest to warrant publication in Nature Communications.

With all due respect to the reviewer, we must disagree. The novelty and impact of our study is now supported by 3 independent reviewers of the manuscript, and goes beyond the fact that our dataset is the first to place isotopic constraints on not one, but several tropical cyclones in a key region of the global tropics. Given that models forced with greenhouse gases suggest that that warming conditions may enhance the frequency of such events in this region, our dataset enables a direct comparison between modern-day TC events and their counterparts in the paleoclimatic records to investigate this scenario using observational constraints that are otherwise unavailable. Our data also provide for future detailed data-model comparisons of isotope-equipped models that simulate TC dynamics, which could help constrain the accuracy of convective parameterizations in climate models.

I also have several specific comments/edits, as follows:

Lines 40 and 46 – “TCs” does not need the “s” here and elsewhere when not plural. **R/Suggested edit completed.**

Line 71 – It is not clear what “latter” is referring to: **R/Sentences was rephrased.**

Line 73 – delete “to”: **R/Suggested edit completed.**

Lines 81-82 – worth also mentioning isotope enabled models that can also be used to investigate water isotope variability in data poor regions. **R/Suggested edit completed.**

Lines 87-90 – some references should be included here: ”: **R/Suggested edit completed.**

Line 92 – “monthly” can be deleted – the amount effect does not specifically refer to monthly values”: **R/Suggested edit completed.**

Line 98 – quantitative transfer functions are rarely used – though if they are, this is a valid critique: ”: **R/Sentences was rephrased.**

Lines 105 and 111 – replace “featured” with “characterized” ”: R/Suggested edit completed.

REVIEWERS' COMMENTS:

Reviewer #3 (Remarks to the Author):

The authors have now satisfactorily addressed my concerns during this latest revision and I am now convinced that this unique dataset will be of broad enough interest to warrant publication in Nature Communications. I look forward to seeing it published.

I did find one minor error on line 107: "qualitative" should be changed to "qualitatively".

Detailed point by point revision

Reviewer #3 (Remarks to the Author):

The authors have now satisfactorily addressed my concerns during this latest revision and I am now convinced that this unique dataset will be of broad enough interest to warrant publication in Nature Communications. I look forward to seeing it published.

R/Thanks for your evaluation and helping us to improve our revised manuscript.

I did find one minor error on line 107: "qualitative" should be changed to "qualitatively".

R/Word was modified.